# The Predictive Capabilities of Artificial Intelligence-Based OCT Analysis for Age-Related Macular Degeneration Progression—A Systematic Review

**DOI:** 10.3390/diagnostics13142464

**Published:** 2023-07-24

**Authors:** George Adrian Muntean, Anca Marginean, Adrian Groza, Ioana Damian, Sara Alexia Roman, Mădălina Claudia Hapca, Maximilian Vlad Muntean, Simona Delia Nicoară

**Affiliations:** 1Department of Ophthalmology, “Iuliu Hatieganu” University of Medicine and Pharmacy, Emergency County Hospital, 400347 Cluj-Napoca, Romania; ioana.damian@umfcluj.ro (I.D.); madalina.prodan06@gmail.com (M.C.H.); stalu@umfcluj.ro (S.D.N.); 2Department of Computer Science, Technical University of Cluj-Napoca, 400114 Cluj-Napoca, Romania; anca.marginean@cs.utcluj.ro (A.M.); adrian.groza@cs.utcluj.ro (A.G.); 3Faculty of Medicine, “Iuliu Hatieganu” University of Medicine and Pharmacy, 400347 Cluj-Napoca, Romania; roman.sara.alexia@elearn.umfcluj.ro; 4Plastic Surgery Department, “Prof. Dr. I. Chiricuta” Institute of Oncology, 400015 Cluj-Napoca, Romania; maximilian.muntean@iocn.ro

**Keywords:** machine learning, deep learning, prediction, age-related macular degeneration, progression

## Abstract

The era of artificial intelligence (AI) has revolutionized our daily lives and AI has become a powerful force that is gradually transforming the field of medicine. Ophthalmology sits at the forefront of this transformation thanks to the effortless acquisition of an abundance of imaging modalities. There has been tremendous work in the field of AI for retinal diseases, with age-related macular degeneration being at the top of the most studied conditions. The purpose of the current systematic review was to identify and evaluate, in terms of strengths and limitations, the articles that apply AI to optical coherence tomography (OCT) images in order to predict the future evolution of age-related macular degeneration (AMD) during its natural history and after treatment in terms of OCT morphological structure and visual function. After a thorough search through seven databases up to 1 January 2022 using the Preferred Reporting Items for Systematic Reviews and Meta-Analyses (PRISMA) guidelines, 1800 records were identified. After screening, 48 articles were selected for full-text retrieval and 19 articles were finally included. From these 19 articles, 4 articles concentrated on predicting the anti-VEGF requirement in neovascular AMD (nAMD), 4 articles focused on predicting anti-VEGF efficacy in nAMD patients, 3 articles predicted the conversion from early or intermediate AMD (iAMD) to nAMD, 1 article predicted the conversion from iAMD to geographic atrophy (GA), 1 article predicted the conversion from iAMD to both nAMD and GA, 3 articles predicted the future growth of GA and 3 articles predicted the future outcome for visual acuity (VA) after anti-VEGF treatment in nAMD patients. Since using AI methods to predict future changes in AMD is only in its initial phase, a systematic review provides the opportunity of setting the context of previous work in this area and can present a starting point for future research.

## 1. Introduction

Age-related macular degeneration (AMD) is the leading cause of irreversible vision loss and legal blindness in individuals over the age of 50. It is currently estimated that, in 2040, approximately 288 million people will be affected by AMD [1]. As a result, eye examinations will increase in frequency, with practices becoming busier and ophthalmologists having less time to analyze their patients’ data. At the same time, the amount of data from one single patient will expand as multimodal imaging continues to evolve and new biomarkers arise. Therefore, it is of great importance for computer-aided algorithms to automate the collection and processing of information. Once processed, these algorithms could help in figuring out the relevant data, leaving doctors more time for human-to-patient interaction.

Individuals in the early and intermediate AMD (iAMD) stages are asymptomatic and the progression to the advanced stage varies between different patients with a varying speed of advancement. Therefore, accurately predicting the patients who will progress is undoubtedly of great importance because the loss of vision starts only in the advanced stage.

The advanced stage of AMD can be further divided into two categories: neovascular AMD (nAMD; also called “exudative” or “wet”) and geographic atrophy (GA; also called “atrophic” or “dry”). Unfortunately, even though data from clinical trials using new emerging therapies seem to indicate that they are favorable in reducing lesion growth, there are no approved therapeutics commercially available for GA [2]. On the other hand, patients suffering from nAMD can benefit from treatment with intravitreal agents against vascular endothelial growth factor (anti-VEGF) and, recently, angiopoietin-2 (anti-Ang-2) [3].

Patients suffering from nAMD in one eye are at increased risk of conversion (progressing from early or intermediate to the late stages of AMD) in their fellow eye, with the risk reaching up to 20% in 2 years [4]. In order to prevent vision loss, the monitoring of the healthy fellow eye should be as important as that of the treated eye.

Optical coherence tomography (OCT) is an easy-to-perform, fast and non-invasive technique that, by means of interferometry, allows visualization of the retinal layers through high-resolution cross-sectional images. Due to its ease of use, OCT is one of the most prevalent ophthalmic imaging modalities, with 30 million scans performed every year [5].

Artificial intelligence (AI) in medicine is currently used either as a support for the clinical decisions or in the imaging analysis. Ophthalmology, as an image-based specialty, is considered to have the widest scope for the application of computer algorithms. The most promising area is the field of retinal diseases, such as AMD, diabetic retinopathy or retinopathy of prematurity, where researchers have developed models for segmentation, classification and, more recently, future prediction [6].

The classical machine learning (ML) models rely on features that are measured directly from the collected data. The model is trained on one part of the dataset, called the training set, and learns to associate certain features with known labels (e.g., classes for classification tasks) or a numerical value (e.g., the predicted value for the regression task). To assess its performance, the model is then evaluated on a different part of the dataset that has not been seen before called the test set. The effectiveness of ML models depends on finding powerful and appropriate prediction models for the given task and feeding them meaningful features for each label.

Deep learning (DL) evolved as a more powerful subset of machine learning [7] and relies on a large variety of architectures for artificial neural networks. Within a deep neural network (DNN), the signal undergoes various transformations during computation as it passes from the first layer, called the input, to the last layer, called the output. As such, these DNNs autonomously learn features from data in a multistep process of pattern recognition without the need for the developers to explicitly identify them beforehand, acting both as a feature extractor and as classifier.

As the field of ophthalmology is image-rich, with fundus photography and OCT commonly integrated in the clinical practice, vast amounts of imaging data are available for AI research. This is why, for the first time, the US Food and Drug Administration (FDA) has approved an autonomous AI-based diagnostic system in ophthalmology that can detect diabetic retinopathy in primary care [8].

In ophthalmology, future prediction tasks powered by AI are mainly focused towards predicting the progression in terms of retinal structure and function after treatment or during the natural evolution of the disease.

In the management of nAMD patients, greater visual acuity (VA) at the initiation of the anti-VEGF therapy is associated with a higher probability of a better visual outcome in the following 2 years under treatment [9,10]. Thus, being able to predict the moment of conversion from early or iAMD to nAMD would be of utmost importance. At the same time, foreseeing the patients who are more at risk would help physicians better schedule their monitoring intervals, catching the conversion sooner and improving treatment outcomes. Predicting the moment of conversion to GA also carries significant value since recent clinical trials show favorable responses in slowing its progression [2].

Predicting the evolution of both nAMD and GA would also be of high value for new developing drugs targeting these types of disease. This would help in proper patient selection for different types of treatment, as well as in evaluating treatment efficacy and establishing valuable trial endpoints. Since anti-VEGF treatment has established itself as the cornerstone in modern nAMD management, predicting the treatment efficiency and its requirements could substantially improve clinical practice, benefiting both patients and ophthalmologists.

An individual’s autonomy and quality of life are greatly dependent on their VA. Among nAMD patients, there is high inter-individual variability in patient responses to anti-VEGF medication in terms of visual function. Therefore, we can see the importance of AI models that could predict future VA. Having the capacity to predict a future increase could help in motivating patients to be compliant and adhere to their demanding follow-up visits, and a future decrease could help them in managing their expectations. The prediction could also help their physicians in considering whether to keep or change their treatment or recommend clinical trials testing upcoming treatments.

## 2. Materials and Methods

### 2.1. Methodology of the Literature Search

In our research, we went through the following steps. First, we defined the research problem based on the following research question: what is the value of machine learning and deep learning OCT analysis for the prediction of the evolution of AMD during its natural history and after treatment in terms of structure and visual function? Second, we found pertinent articles that met the pre-set inclusion criteria. Third, we extracted the relevant data from the selected articles. Fourth, we assessed the quality of the extracted data. The current systematic review was carried out through meticulous and exhaustive research of the following seven databases (all the links provided below were accessed on 22 September 2022).
ArXiv18 articles(https://arxiv.org/)Cochrane Library18 articles(https://www.cochranelibrary.com/)Embase258 articles(https://www.embase.com)IEEE Xplore29 articles(https://ieeexplore.ieee.org)Pubmed113 articles(https://pubmed.ncbi.nlm.nih.gov/)Science Direct362 articles(https://www.sciencedirect.com)Scopus1002 articles(https://www.scopus.com/)

The systematic review was carried out following the 2020 Preferred Reporting Items for Systematic Reviews and Meta-Analyses (PRISMA) guidelines [11]. Due to the unsatisfactory reporting in systematic reviews [12], the PRISMA statement was intended to serve as a reporting guideline, ensuring that reviewers conduct their work in a transparent manner permitting their readers to fully understand their reasonings, means and findings. We searched seven databases for all articles published before 1 January 2022 and this resulted in a total of 1800 records being retrieved and uploaded in the reference manager software Mendeley. After removing 396 duplicate records using the Mendeley software, 1404 records remained for screening. The 1404 records were split in half and two authors (G.A.M. and S.A.R) independently screened the titles and abstracts of the records for each half. This process led to the exclusion of 1356 records. Following exclusion, 48 full-text articles were retrieved and assessed for eligibility independently by the same two authors (G.A.M. and S.A.R). In cases of disagreement, consensus regarding inclusion/exclusion was reached through discussions and, if necessary, a third author (A.M.) was consulted. In this review, we included all journal papers and conference papers found in the previously mentioned databases using the following keywords for searching: “machine learning” or “deep learning” and “age-related macular degeneration” and “OCT”. The PRISMA flow chart is displayed in Figure 1.

### 2.2. Inclusion Criteria

The goal of our research was to find, chart and analyze the articles exploring the predictive value of ML and DL OCT analysis for studying AMD’s natural history and therapeutic response. We therefore included those studies that analyzed OCT images with ML and DL algorithms to predict AMD’s evolution in terms of structure and visual function during the disease’s natural history or in response to treatment.

We included articles that analyzed:The progression of AMD from early and iAMD stages to nAMD and GA;The progression of nAMD under anti-VEGF therapy;The progression of GA in its natural history and under new trial treatments;The efficacy of anti-VEGF treatment of nAMD;The requirements of anti-VEGF treatment of nAMD;The VA outcomes after anti-VEGF treatment of nAMD.

We included all articles found suitable that were published up to 1 January 2022.

### 2.3. Exclusion Criteria

We excluded the following type of articles:Articles that used ML and DL algorithms but based their prediction on color fundus photographs and other types of imaging (e.g., confocal scanning laser ophthalmoscopy) and did not include OCT analysis;Articles that analyzed OCT images with ML and DL algorithms but focused on learning tasks, such as classification or segmentation of AMD;Articles that analyzed OCT images with ML and DL models but used statistical analysis for their prediction models.

### 2.4. Selection of Papers

We selected those articles that met our inclusion criteria. Following this triage, we ended up with 19 articles from which we built up our review. We saved each article as a pdf document and used the title to name it.

## 3. Results

We finally arrived at the 19 articles displayed in Table 1 that satisfied our inclusion criteria. The datasets utilized by the articles are presented as number issued per year in Figure 2 and as number issued per country of origin in Figure 3. Among these 19 studies:Four articles predicted the need for anti-VEGF treatment among nAMD patients; three used ML models and one used a DL model;Four articles predicted anti-VEGF treatment efficacy for nAMD patients; all four used DL models and, among them, two used generative adversarial networks (GANs);Four articles predicted the conversion from early or iAMD to nAMD; one used a ML model and three used DL models;Two articles predicted the conversion from iAMD to GA; one used a ML model and one used a DL model;Three articles predicted the growth of GA; one used a ML model and two used DL models;Three articles predicted the VA outcome after anti-VEGF treatment in nAMD patients; two used ML models and one used a DL model.

Since Schmidt-Erfurth et al. [13] developed a ML model for predicting the conversion to both nAMD and GA, the findings for both types of predictions will be described in Section 3.4.
diagnostics-13-02464-t001_Table 1Table 1The 19 articles included in the review arranged by prediction category and colored according to the type of model used (green = ML; blue = DL).Prediction201620172018201920202021Treatment requirements
Bogunovic et al. [14]

Romo-Bucheli et al. [15]Pfau et al. [16]Gallardo et al. [17]Treatment efficacy



Feng et al. [18];Lee et al. [19];Liu et al. [20]Zhao et al. [21]Conversion to nAMD

Schmidt-Erfurth et al. [13]Russakoff et al. [22]Yim et al. [23];
Banerjee et al. [24]Conversion to GA

Schmidt-Erfurth et al. [13]Rivail et al. [25]

GA growthNiu et al. [26]


Zhang et al. [27];
Gigon et al. [28]VA outcome

Schmidt-Erfurth et al. [29]
Kawczynski et al. [30]
Rohm et al. [31]

### 3.1. Prediction of Treatment Requirements—Classic Machine Learning

Bogunovic et al. [14] used a random forest classifier to predict low and high anti-VEGF treatment requirements for 317 eyes from 317 patients undergoing a 2 year pro re nata (PRN) schedule in the phase-three HARBOR clinical trial during their initiation phase (baseline, month one and month two). The HARBOR clinical trial included patients with treatment-naive subfoveal choroidal neovascularization (CNV) secondary to nAMD. During the trial, one eye per patient was randomized in a 1:1:1:1 fashion to Ranibizumab with the following scenarios: (1) 0.5 mg monthly, (2) 0.5 mg PRN, (3) 2.0 mg monthly and (4) 2.0 mg PRN. In the PRN schedule, patients had a course of three Ranibizumab injections in the first 3 months and were then followed-up monthly with Cirrus HD-OCT III imaging and VA measurements for retreatment criteria (a five-letter drop in best-corrected visual acuity (BCVA) compared to the previous visit or any evidence in follow-up OCT images of disease activity). Patients receiving ≤5 anti-VEGF injections were marked as low-treatment, those receiving ≥16 as high-treatment and anything between as medium-treatment. From the baseline, month one and two visits, the model used eight OCT features for each of the 13 regions (nine Early Treatment Diabetic Retinopathy Study (ETDRS) regions + three additional ones): four retinal-layer 2D map thicknesses derived from graph theory automatic segmentation and volume and en face area maps for intraretinal fluid (IRF) and subretinal fluid (SRF) obtained from convolutional neural network (CNN) segmentation. Temporal differences between features, BCVA, demographic characteristics and supplementary clinical and imaging data were included such that the final model had 530 features: ((8 feature maps × 13 regions + 1 BCVA) × 5 temporal elements) + sex, age, race, smoking status and fluorescein angiography (FA) pattern type. The values for the area under the receiver operating characteristic curve (AUC) were 0.7 and 0.77 for the low versus others and high versus others treatment requirements. The top three most important features were: (1) SRF volume in the central 3 mm circle at month two; (2) inner retina thickness at the fovea at month one; (3) inner retina thickness in the central 3 mm circle at month two. Interestingly, the authors observed that temporally differential features seemed to have a similar role as features observed in a cross-sectional analysis in the model’s prediction. The least important features were sex, FA pattern type, smoking status and race.
Figure 2The number of datasets per year from the 19 articles included in the current systematic review.
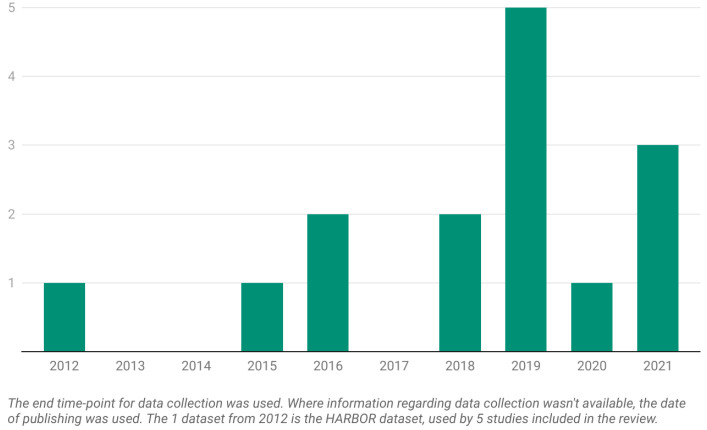



Pfau et al. [16] evaluated the ability of a probabilistic natural gradient boosting (NGBoost) forecasting model to predict future anti-VEGF treatment requirements using biomarkers extracted from volumetric spectral domain OCTs (SD-OCTs). In contrast to Bogunovic et al. [14], who provided the prediction point for each patient but without a full probability distribution over the entire outcome space, the model developed by Pfau et al. [16] provided a measure of predictive uncertainty for each individual prediction. The dataset included real-world data (RWD) from 40 visits for 40 eyes belonging to 37 patients with nAMD in at least one eye visiting the Department of Ophthalmology and Visual Sciences, University of Illinois, Chicago, IL, USA, together with data from 108 visits for 59 eyes belonging to 59 patients visiting the University Eye Hospital Bonn, University of Bonn, Germany. Patients were treated with different anti-VEGF inhibitors (Bevacizumab, Ranibizumab or Aflibercept) with either a PRN or treat and extend (T&E) protocol. Both treatment-naive and pre-treated patients at the time of the SD-OCT scan were included. The imaging protocol consisted of 20° × 15° SD-OCT imaging (19 B-scans, ART 8) using a Spectralis HRA+OCT device (Heidelberg Engineering, Heidelberg, Germany). The SD-OCT B-scans were segmented using a previously trained and validated CNN (Deeplabv3 model with a ResNet-50 backbone). Retinal thickness (mean and standard deviation (SD)) and layer reflectivity values (minimum-, mean- and maximum-intensity projections) were extracted for the central and the four inner ETDRS subfields for the following segmented layers: inner retina, outer nuclear layer (ONL), inner segment (IS), outer segment (OS), retinal pigment epithelium-drusen complex (RPEDC) and choroid. The total number of features equaled 270 (27 maps × 2 measures (mean/SD) × 5 (ETDRS subfields)). Three conventional machine learning algorithms applicable in the setting of correlated predictors were probed in this study in order to predict the anti-VEGF treatment frequency within the following 12 months: 1. lasso regression, 2. principal component regression and 3. random forest regression. For probabilistic forecasting, the authors implemented a natural gradient boost (NGBoost) with a negative binomial distribution outputting a full probability distribution. The primary outcome measure in the present study was the mean absolute error (MAE) for the predicted versus the actual anti-VEGF treatment frequency. Only one visit and one eye per patient were included in the model fitting. The predictions of the anti-VEGF treatment frequency, with accuracy in terms of the MAE and 95% CI, were 2.76 injections/year (2.39–3.14) ($R^{2}$ = 0.038) using lasso regression and 2.74 injections/year (2.38–3.11) ($R^{2}$ = 0.173) using principal component regression. For the random forest regression, the provided prediction accuracy was 2.60 injections/y (2.25–2.96) ($R^{2}$ = 0.390). The probabilistic prediction with NGBoost was similarly accurate: 2.66 injections/year (2.31–3.01) ($R^{2}$ = 0.094). The AUCs for the low treatment requirement (defined as ≤4 injections/year) and high treatment requirement (defined as ≥10 injections/year or more) were as follows: lasso regression—0.61 and 0.63, principal component regression—0.63 and 0.7, random forest regression—0.68 and 0.7 and NGBoost—0.68 and 0.69. The models’ performances are displayed in Table 2. The SD for RPEDC thickness in the central ETDRS subfield was found to be an important predictor across models, meaning that higher values for the SD for the central RPEDC thickness were associated with a greater need for injections. In terms of estimations, all models predicted more injections than necessary for low treatment requirements and fewer than necessary for high treatment requirements. The present study was limited by the sample size, as well as by the different disease phenotypes not included in the training data.
diagnostics-13-02464-t002_Table 2Table 2Pfau et al. [16]: conventional machine learning models vs. NGBoost for predicting anti-VEGF treatment requirements during the following 12 months (AUC = area under the curve; inj. = injections; Tx = treatment).
Lasso RegressionPrincipal Component RegressionRandom Forest RegressionNGBoostMAE for no. of inj. for 12 months (95% CI)2.76 (2.39–3.14)  (R2 = 0.038)2.74 (2.38–3.11) (R2 = 0.173)2.60 (2.25–2.96)  (R2 = 0.390)2.66 (2.31–3.01)  (R2 = 0.094)AUC for low Tx requirement (≤4 inj.)0.610.630.680.68AUC for high Tx requirement (≥10 inj.)0.630.70.70.69
Figure 3The number of datasets per country from the 19 articles included in the current systematic review (United States—5, China—4, Germany—2, Switzerland—2, United Kingdom—2, South Korea—1; map is scaled).
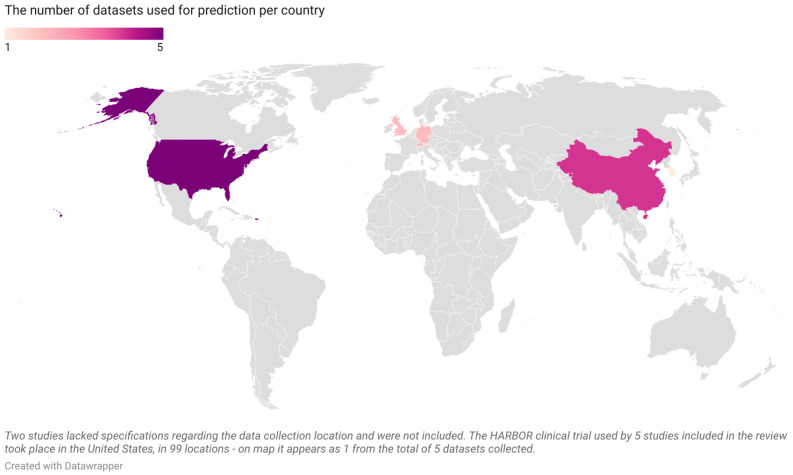



Gallardo et al. [17] studied ML models’ capacities to classify treatment needs with 710 eyes from 625 patients from the University Hospital of Bern who underwent anti-VEGF therapy in a T&E fashion. Among the 710 eyes included, 377 eyes had nAMD (340 patients), 155 eyes had macular edema-related retinal vein occlusion (RVO) (150 patients) and 178 eyes had diabetic macular edema (DME) (135 patients). The patients included in the study underwent at least 365 days of T&E with either Ranibizumab or Aflibercept between 1 January 2014 and 31 December 2018. The macular 6 × 6 mm OCT volumes were acquired using the Heidelberg Spectralis SD-OCT imaging system. The treatment demand was established for the first year of treatment as low, moderate or high. The cutoff value for treatment intervals for the low-treatment group was ≥10 weeks or more, and for the high-treatment group, it was ≤5 weeks, while everything in between was attributed to the moderate-treatment group. Two binary random forest classifiers were built with 1000 trees with a maximum tree depth of 100 to differentiate between the low-demand group versus others (LvO) and the high-demand group versus others (HvO). The models used both morphological features extracted from the OCTs as well as the patient’s sex and age at each visit. The models used 131 segmentation-based morphological features ((4 groups of layers × 1 mean thickness × 13 ETDRS regions) + (3 fluids × 2 en face area map and volume features × 13 ETDRS regions) + 1 patient-independent retinal thickness). These were coupled with biomarker-based morphological features detected with a CNN classifier for each individual B-scan (10 total features: SRF, IRF, hyperreflective foci (HRF), drusen, reticular pseudodrusen, epiretinal membrane, GA, outer retinal atrophy and fibrovascular pigment epithelial detachment (PED)). Two types of features were built for each of the 10 biomarkers: (1) the number of B-scans, where the probability for their presence was ≥0.75, and (2) the maximum probability for their presence across all B-scans. The models also used differential features, where they measured the differences in morphological features over time from visit one to baseline and visit two to visit one. After combining the features ((131 (segmentation-based) + 20 (detection-based)) × 3 (baseline, V1 and V2) + 604 (differential features) + 1 (sex) + 3 (age at each visit)), the result equaled a total of 1061 features. The authors performed 10-fold cross-validation, with 90% of the data in the train set and 10% in the test set, evaluated by the mean AUC for all 10 folds, which was 0.79 for both the LvO and HvO nAMD classifiers for the 1-year treatment demand, taking into consideration the baseline + first two consecutive visits. They computed the feature importance both in terms of statistical significance and relevance for the model’s prediction. The authors concluded that, in order to classify the low-demand group, features extracted from the first visit were enough, as opposed to the high-demand group, for which features from the second and third visits were necessary.

### 3.2. Prediction of Treatment Requirements—Deep Learning

Romo-Bucheli et al. [15] developed an end-to-end DL model that aimed at predicting low and high treatment requirements for AMD patients from longitudinal retinal OCT images. The model’s architecture is built on three components. First, a DenseNet network is used to extract the features from the OCT images. Second, a recurrent neural network (RNN) integrates the information obtained from the OCT images from multiple time points within the initiation phase. Last, a fully connected layer integrates spatiotemporal information from the RNN and outputs the probability for a patient to be a part of each treatment requirement group (low, intermediate and high). The categories were defined as follows: low—≤5 injections, high—≥16 injections and intermediate—everything in between. From the total number of 350 patients, the authors used 70% of the data (247 patients) for training, 10% (34 patients) for validation and 20% (69 patients) for testing. For the three-class classification task (low, intermediate and high), the model yielded the following accuracy, sensitivity and specificity: (0.72, 0.82, 0.69) for the low-treatment group, (0.65, 0.61, 0.71) for the intermediate-treatment group and (0.9, 0.5, 1.0) for the high-treatment group. The model was also tested on two additional tasks: (1) two binary classifications (high-treatment group vs. all the rest and low-treatment group vs. all the rest), where the model was compared to a baseline ML model similar to the model used by Bogunovic et al. in [14]; and (2) a regression task to predict a treatment requirement score (the number of received injections divided by the total number of visits, without taking into account the initiation phase). The high-treatment group vs. others classification yielded an AUC of 0.81 while the low-treatment group vs. others classification yielded an AUC of 0.85. The regression model obtained a Pearson correlation coefficient *R* = 0.59 and a coefficient of determination $R^{2}$= 0.22. They finally evaluated the model’s decision using an adapted occlusion sensitivity method. Even though they evaluated the patient-specific heatmaps, the authors generated a representative attribution heatmap for each treatment requirement category displaying regions that were more important for the model’s decision. For the high and intermediate groups, the heatmap was scattered along the image, while for the low group, it was concentrated in a small zone below Bruch’s membrane. When comparing the low- and high-treatment groups, the heatmaps seemed to be complementary, with regions more important in the low category having less importance in the high category and vice versa.

### 3.3. Prediction of nAMD Anti-VEGF Treatment Efficacy—Deep Learning

Feng et al. [18] trained a CNN-based predictive model to assess the efficacy of intravitreal anti-VEGF injections in patients with two types of macular lesions: CNV and cystoid macular oedema (CME). CNV is mainly secondary to nAMD and CME is secondary to diabetic retinopathy, but both can also appear as part of other retinal diseases. The data consisted of 228 OCT images from 228 patients diagnosed with CNV, CME or both lesions simultaneously who were treated at the Second Affiliated Hospital of Xi’an Jiaotong University between 2017 and 2019. An OCT was recorded prior to the anti-VEGF injection and the evaluation of the treatment’s efficacy was determined after 21 days (for 171 patients (75%), the response was effective, while for the 57 remaining patients (25%), there was no sign of effectiveness). The criteria for appreciating effectiveness were not described in the paper. Before training the CNN, the OCT images underwent preprocessing and the dataset was split randomly into training (80%) and testing (20%) for each class. This was followed by data augmentation through various techniques, resulting in a total number of 912 OCT images. The chosen model for the prediction was a modified ResNet-50 pre-trained on ImageNet, which was compared to other known architectures, like AlexNet, VGG-16 and GoogLeNet. The ResNet-50 model was trained with four datasets: (1) a set containing the full OCT image with data augmentation (full w DA), (2) a set containing the full OCT image without data augmentation (full w/o DA), (3) a set containing only the lesion from the region of interest with data augmentation (lesion w DA) and (4) a set containing only the lesion from the region of interest without data augmentation (lesion w/o DA). The resulting AUC, accuracy, sensitivity and specificity are described in Table 3. When the performance of the four CNN architectures mentioned before was compared on the full w DA dataset, ResNet-50 achieved the highest AUC (0.81), followed by GoogLeNet and VGG16 (0.75) and AlexNet (0.72).

Liu et al. [20] harnessed the power of GANs and used them to predict the short-term response following intravitreal anti-VEGF treatment of nAMD. The dataset consisted of patients treated at Peking Union Medical College Hospital from November 2018 to June 2019. The GAN model was made up of two components: one that generated the images (generative model) and another one that improved them (discriminative model). Based on learning the corresponding 476 pre-therapeutic and post-therapeutic OCT images, the GAN could synthesize a post-treatment OCT image for the 50 pre-therapeutic test images showing the therapeutic effect. An advantage of this approach is that it eliminates the need for labels, segmentation or clinical information, using solely the images as input and generating other images as output. However, this is also a double-edged sword, as their absence pushes us further away from explainability. In their experiments, the retina specialists had a hard time distinguishing the real images from the synthetic ones, meaning that they resembled each other in quality. The main aim of the study was to assess the capacity of the post-therapeutic synthetic images to describe the macular status as either dry or wet and the conversion from wet to dry. The authors defined a dry macula as the absence of SRF and IRF on the OCT image, while the presence of either one defined a wet macula. The prediction accuracy for macular status as either wet or dry was 0.85 (95% CI: 0.74 to 0.95; 84% for wet and 86% for dry status), with 0.84 (95% CI: 0.63 to 0.95) sensitivity, 0.86 (95% CI: 0.63 to 0.96) specificity, 0.88 positive predictive value (PPV) and 0.82 negative predictive value (NPV). The prediction accuracy for wet-to-dry conversion was 0.81 (95% CI: 0.69 to 0.93), with 0.83 (95% CI: 0.58 to 0.96) sensitivity, 0.79 (95% CI: 0.57 to 0.92) specificity, 0.75 PPV and 0.86 NPV.

Lee et al. [19] trained a conditional GAN (cGAN) to generate 1-month-post-loading phase OCT B-scans in order to predict the presence of abnormal elements, such as IRF, SRF, PED and subretinal hyperreflective material (SHRM). The cGAN was trained on 927 B-scans from 309 eyes of 298 nAMD patients treated with at least three anti-VEGF injections (Ranibizumab and Aflibercept) at Konkuk University Medical Center between 2010 and 2019. Besides the OCT images obtained at baseline and 30 days (1 month) after the loading phase, the dataset contained 732 images of both FA and indocyanine green angiography (ICGA) acquired at 5 min post-dye injection. The model was evaluated by comparing the lesions (IRF, SRF, PED and SHRM) present on the predicted cGAN OCT images versus those present on the actual OCT images. Two clinicians and two graders evaluated the presence of the aforementioned lesions on both the real and post-therapeutic OCT images. An interesting finding was that the accuracy, specificity and negative predictive value of the aforementioned elements improved when FA and ICGA images were added to the input. The accuracy of predicting the evolution of lesions based on OCT only as compared to using OCT + FA + ICG was as follows: IRF (89.6% and 92.6%), SRF (77.0% and 80.7%), PED (77.0% and 80.7%) and SHRM (91.9% and 96.3%).

Zhao et al. [21] developed a DL model to predict the response in terms of BCVA evolution after nAMD treatment with anti-VEGF therapy. The dataset used contained 4944 OCT images acquired with the Deep Range Imaging (DRI) OCT Triton device with the follow-up mode from 206 eyes of 181 nAMD patients who were treated at Peking Union Medical College Hospital between November 2018 and July 2020. For each OCT image, BCVA in the Snellen format was recorded. Patients underwent the initiation phase with three monthly anti-VEGF injections (Ranibizumab or Aflibercept) followed by the PRN regimen. Based on BCVA evolution after anti-VEGF injections, patients were grouped as responders (improvement of one Snellen line or more) and non-responders (reduction or stabilization). The training set was built with 71 volume pairs from responders and 89 from non-responders, the validation set with 10 from responders and 10 from non-responders and the test set with 13 from responders and 13 from non-responders. Pre-processing steps included cropping the region of interest (ROI), image alignment and data augmentation. The framework consisted of two parts—training and inference. For the training phase, the classifier together with the sensitive structure locator made a U-net architecture, where they acted as an encoder and decoder. In this phase, the sensitive structure-guided network (SSG-Net) model used for prediction improved the capacity to classify patients as responders or non-responders by paying attention to the structural differences between pre-injection and post-injection images. The Squeeze-and-Excitation ResNet (SE-ResNet) blocks acted as the encoder, connected by global average pooling and fully connected layers to output the result. A sensitive structure locator acted as a decoder with a top-down architecture with lateral connections and captured the variation tendencies between pre- and post-injection OCT images. In the inference phase, the classifier used only the pre-treatment image for its final prediction, classifying patients as responders and non-responders. The SSG-Net model’s prediction was evaluated in contrast to other models in a machine–machine comparison and to humans in a human–machine comparison. The machine–machine comparison evaluated the performance of SSG-Net with SE-ResNet-50 in two scenarios: (1) SE-ResNet-50 with only the pre-injection OCT image and (2) SE-ResNet-50 with the pre-injection OCT image plus the post-injection synthesized OCT image. SSG-Net achieved 0.83 AUC vs. 0.74 for SE-ResNet-50 in scenario two and 0.66 for SE-ResNet-50 in scenario one, 84.6% accuracy vs. 73.1% and 65.4%, 0.692 sensitivity vs. 0.692 and 0.461 and specificity of 1 vs. 0.846 and 0.846, respectively. The four physicians’ performance evaluated by confusion matrices yielded the following results: accuracy of 76.9%, 53.8%, 69.2% and 53.8%; sensitivity of 0.923, 0.538, 0.538 and 0.692; and specificity of 0.615, 0.538, 0.846 and 0.385.

### 3.4. Prediction of Conversion to nAMD and GA—Classic Machine Learning

Schmidt-Erfurth et al. [13] developed a complex ML model involving OCT biomarkers alongside demographic and genetic features to predict the conversion from iAMD to nAMD and GA in the following 2 years in fellow eyes of patients treated for nAMD. The dataset contained imaging and clinical information from the 495 fellow untreated eyes of 495 patients enrolled in the HARBOR clinical trial. The moment of conversion to either nAMD or GA appearing on longitudinal OCT images was manually decided by two graders. In the automated image analysis step, 2D en face thickness maps were generated from three retinal layers (ONL, retinal pigment epithelium (RPE) + IS/OS and space occupied by drusen between the RPE and Bruch’s membrane), drusen + pseudodrusen and HRF. From these 2D thickness maps, the authors automatically extracted the following biomarkers: 8 measurements derived from drusen, 2 from pseudodrusen, 15 from the three retinal layers and 9 from the HRF. The mean values of the imaging biomarkers measured at baseline and their changes during the four next follow-up visits, together with demographic (age, sex), smoking and genetic (34 single-nucleotide polymorphisms (SNPs) associated with AMD) status, served as input for a sparse Cox proportional hazards (CPH) model. Fed with these features, the model output a hazard ratio (HR) that was used as the risk for eye conversion. The model was evaluated in a 10-fold cross-validation. The 2 year AUC for nAMD conversion prediction was 0.68 (with 0.46 specificity for 0.80 sensitivity) and the AUC for GA conversion prediction was 0.80 (0.69 specificity for 0.80 sensitivity). The revelation was that the model spotted different patterns for eyes converting to nAMD versus those developing GA, thus reinforcing the idea that they have distinct pathways of evolution. The most important features for nAMD conversion were drusen-centric and included thickening of the RPE–drusen complex, increases in the drusen area, drusen-centric HRF and thickening of the ONL in regions of HRF concentration. For GA conversion, the most important features were retina-wide and were related to outer retinal thinning: higher variability in the RPE + IS/OS thicknesses, decreases in the RPE + IS/OS minimal thickness, decreases in the ONL thickness and increased HRF in the ONL layer. It is worth mentioning that, for the non-imaging features, the genetic profiles did not show prognostic value in predicting GA and nAMD conversion, and only age appeared to influence the prognosis for GA development.

### 3.5. Prediction of Conversion to nAMD—Deep Learning

Russakoff et al. [22] compared the performance of two deep-learning CNNs, one called AMDnet, which was trained from scratch, and the other being an already popular CNN image recognition model, VGG16 [32], which was fine tuned with transfer learning. The dataset included 2-year longitudinal OCT images from 71 fellow eyes with early or iAMD from 71 patients with nAMD in the contralateral eye treated with anti-VEGF at Moorfields Eye Hospital. Each patient had three OCT images acquired at baseline and at the end of each of the following 2 years. Both CNN models were compared with and without image preprocessing in the task of correctly classifying the progressors from early and iAMD to advanced AMD using only OCT imaging. For both models, the preprocessing step significantly improved their performance, with AMDnet reaching an AUC of 0.89 on OCT B-scans and 0.91 on OCT volumes (represented as the mean from all B-scans in the volume), while VGG16 achieved 0.82 and 0.87, respectively. The authors concluded that accounting for different axial resolutions resulting from different OCT devices’ spectrometers does not improve the model’s performance. Also, small differences in follow-up lengths did not seem to introduce significant bias in the results. Important information was revealed from feature analysis, which showed certain OCT structures most likely to influence the classifier’s prognosis. After performing an occlusion sensitivity analysis at the B-scan level, pixels around the RPE seemed to have the greatest influence on the final score for those who were not progressing. Meanwhile, for those progressing, pixels under the RPE or involving the choroid seemed to be more important. The authors tried to compare the model’s prediction with three retinal specialists, but they refused the task, even in the setting of a research study, because of the lack of biomarkers to rely upon. The authors thought that, behind its prediction, the model looked at these important OCT regions and found specific patterns that pointed out pathological changes. As the observed regions did not yet contain visible pathological elements, more subtle features were presumably being identified, such as changes in textures.

Banerjee et al. [24] developed a hybrid sequential prediction model named “Deep Sequence” using two-layer one-directional stacked stateful long short-term memory (LSTM) units. This model uses quantitative spatiotemporal features automatically extracted from longitudinal OCT images, demographic information (age, gender, race and smoking status) and BCVA, which are fed to an RNN model. The features comprise 21 imaging features, which describe the presence of drusen and their number, extent, density and relative reflectivity. The RNN model then predicts the onset of an exudative event (nAMD) in eyes with early or iAMD in different time spans, from 3 up to 21 months, using the previous visit history. The training was carried out through 10-fold cross-validation, using 671 early or iAMD fellow eyes from 671 nAMD patients undergoing anti-VEGF treatment within the HARBOR clinical trial. This resulted in excellent prediction for 3 months, with an AUC of 0.96 ± 0.02, and good prediction for 6 months, with an AUC of 0.83 ± 0.04. The model’s performance dropped for 12 months (0.77 ± 0.06) but recovered for 18 months (0.9 ± 0.06) and 21 months (0.97 ± 0.02). The model’s generalizability was validated with an external RWD dataset using 719 early or iAMD eyes from 507 patients visiting Bascom Palmer Eye Institute between 2004 to 2015. It achieved high performance with an AUC = 0.82 for the 3-month short-term prediction but weaker prediction with increasing time frames: 6 months—0.77, 12 months—0.69, 18 months—0.65, 21 months—0.68. The performance drop might have been due to the following reasons. There was a wide distributional shift between the two datasets: at 21 months, the number of progressors in the HARBOR dataset was higher at 41% than in the MIAMI dataset. This was linked to the fact that the patients in the HARBOR trial already had a contralateral eye with nAMD, thus making them more prone to nAMD in the fellow tested eye. Another aspect was the fact that smoking status and ethnicity were not available in the MIAMI dataset and were thus considered missing data. The model was also tested with different visit numbers and the authors observed that the Deep Sequence model improved when the number of historic visits was increased.

Yim et al. [23] developed a deep learning prediction model built with two components to predict the moment of conversion to nAMD within a 6-month window frame using a single OCT scan. The study investigated patients with nAMD in one eye, and the fellow eye was the one analyzed for the conversion. The first component was made up of a two-stage architecture that first applies retinal segmentation with the help of a 3D U-Net; afterwards, with this information, a classification network predicts the risk of conversion. The second component makes the same prediction using the raw OCT image alone. The final prediction model is an ensemble of the two mentioned components. The models were trained and tested on 2261 non-AMD fellow eyes of 2795 nAMD patients who visited seven different sites of Moorfields Eye Hospital between 2012 and 2018, with patient level split into a training and validation set (80% of the data) and a test set (20%). The extracted features comprised tissue maps and volumes for 13 tissue classes (vitreous and subhyaloid, posterior hyaloid, epiretinal membrane, neurosensory retina, IRF, SRF, SRHM, HRF, RPE, drusenoid, serous and fibrovascular PED, choroid and outer layers) and three artifacts (mirror, clipping and blink arterfacts). As the patients belonged to a retrospective cohort, the authors used two different time points for conversion: (1) the OCT imaging once patients started intravitreal anti-VEGF injections (“injection scan”) and (2) the exact moment of conversion as detected by expert review of the consecutive OCT images (“conversion scan”). The final model reached an AUC of 0.745 on the test set for the OCT volume conversion scan and an AUC of 0.884 for the OCT volume injection scan. The authors proposed two operating points for the model, one called conservative, where the specificity was 90% and sensitivity 34%, and the other liberal, where sensitivity was 80% and specificity 55%. With the conservative operating point, the false positives appeared in only 9.6% of OCT scans versus 43.4% with the liberal operating point. The model was compared to six human experts (three retina specialists and three optometrists). When using a single OCT scan for predicting conversion, the system outperformed five experts and matched one optometrist. When the human experts had additional OCT historic data, fundus images and demographic and BCVA data, the model still had a higher performance than five experts and matched one retinal specialist, even though it was still using only the single OCT scan. The authors also looked at feature importance and the model’s sensitivity substantially improved when the following extracted features were present: HRF, high drusen volumes, PED.

### 3.6. Prediction of Conversion to GA—Deep Learning

Rivail et al. [25] trained a deep Siamese network in a self-supervised manner with auxiliary tasks, such as predicting the intervals between successive OCTs. This proposed task of estimating time intervals between pairs of images from the same patient was used as a pretext, and the network was focused on learning temporally specific patterns that could be later on used for disease prediction with the help of transfer learning. As this model was trained in a self-supervised manner, it did not rely on annotations or regular sampling intervals and, as such, it can be used with large sets of unlabeled longitudinal data, removing the need for time-consuming manual tasks. The training dataset consisted of 3308 OCT scans from 221 patients of 420 eyes with iAMD. The dataset was divided into six fixed folds for cross-validation. As a preprocessing step, OCT images were aligned using Bruch’s membrane, cropped and resampled. Using only the central OCT B-scan from the oldest acquisition time point before conversion (only one visit), the deep Siamese network reached AUCs of 0.753 ± 0.061, 0.784 ± 0.067 and 0.773 ± 0.074 in predicting the conversion from iAMD to GA within 6, 12 and 18 months.

### 3.7. Prediction of GA Growth—Classic Machine Learning

Paving the way, Niu et al. [26] built an ML model that could forecast the future growth of GA, indicating the direction and speed of the lesion’s spread. The study was based on 38 GA eyes from 29 patients who visited Byers Eye Institute at Stanford University and the features used were extracted automatically from 118 SD OCT scans acquired with Cirrus OCT. The GA lesions from the longitudinal follow-up OCT series were aligned using registration based on blood vessel projection images. The framework consisted of automatic segmentation of the GA lesions from the OCT scans, followed by automatic extraction of 19 quantitative features from each axial scan location. Among the 19 features, 6 described the size and shape of the GA lesions, 3 defined drusen locations and height and reticular pseudodrusen thickness obtained from segmentation and quantification, 1 described the increased minimum retinal intensity, 1 indicated the photoreceptor loss map, 5 accounted for the axial thickness of different layers and the last 3 outlined the mean axial reflectivity between retinal boundaries. These features were fed to a random forest classifier using 100 decision trees to predict if a pixel in a topographic image would develop GA or not at a given time. The model was evaluated in three different scenarios: (1) predicting the GA growth at each patient’s first follow-up visit (using features from the baseline GA and growth at the first follow-up visits for all other patients); (2) predicting the time course for GA growth across all consecutive visits for each patient (using the same features as in scenario one); and (3) predicting the time course for GA growth at each patient’s second follow-up and subsequent visits (using features from the baseline GA and growth at the first follow-up scan, but this time only from the same patient). The most important features in predicting the location of GA growth turned out to be the regions with photoreceptor loss, lower reflectivity in the ellipsoid zone and decreased thickness in reticular pseudodrusen. The authors calculated the mean Dice index (DI) +/− SD for all scenarios between the predicted and observed GA regions (shown in Table 4) for overall GA lesions and for GA growth only: (1) scenario one—0.81 +/− 0.12 for overall GA lesions and 0.72 +/− 0.18 for GA growth only; (2) scenario two—0.84 +/− 0.10 and 0.74 +/− 0.17; (3) scenario three—0.87 +/− 0.06 and 0.72 +/− 0.22.

### 3.8. Prediction of GA Growth—Deep Learning

Zhang et al. [27] approached the task of predicting the growth of GA differently by using a bi-directional long short-term memory (BiLSTM) network with a 3D-UNet CNN refinement. When compared with the previous paper by Niu et al. [26], the novelty came from using two sequential follow-up visits instead of only one to predict the next visit and from converting the different time intervals into time factors and integrating them in the BiLSTM, which then permitted predicting the GA growth at a given time. The dataset contained 22 GA eyes from 22 patients visiting Byers Eye Institute of Stanford University and 3 GA eyes from 3 patients visiting Jiangsu Provincial People’s Hospital in China. As a pre-processing step, layer segmentation, image registration and GA segmentation were applied. The prediction results provided by the BiLSTM model for the three visits (baseline, visit one—known visits—and visit two—speculated visits) and the simulated GA growth maps during these three visits were fed to the 3D-UNet to output the final location of GA growth. The paper described 10 different scenarios, which were compared to highlight important aspects of the learning process. The first six scenarios, detailed in Table 5, looked into whether the law of GA growth was common among the independent patients and evaluated the impact of prior information volume on prediction accuracy. The DI was higher for the patient-dependent scenarios, meaning that even though there was a common pattern in the growth between different patients, the inherent patient specificity limited the patient-independent test’s accuracy. This led to the conclusion that, in predicting GA’s growth, it is more helpful to have similar prior information. When the follow-up series was longer, the prediction accuracy improved, which showed that the network had more powerful learning given more similar follow-ups. The seventh scenario tested the importance of time factors. Leaving them out resulted in a decrease by 10% in the model’s DI. The authors came to the conclusion that, using the time factors, the network learned the unitized growth rate and this helped in its optimization. In the eighth scenario, they compared a CNN-based refinement strategy and a simple voting decision strategy. As expected, the former outperformed in all cases except one where there was not enough data for training. In the ninth scenario, they tested whether two sequential follow-up visits outweighed one single follow-up visit. In line with our expectations, the former’s prediction accuracy was higher. In the last scenario, they tried to estimate the model’s generalization across different geographic regions (USA and China). They found that their model’s limitation could have been due to two main reasons: first, the possibility of imprecise pre-processing by the automated retinal layer and GA segmentation and the automated image registration between follow-up visits; and, second, the limited amount of data, even though augmentation steps were in place.

Gigon et al. [28] trained a DL model to predict future RPE and outer retina atrophy (RORA) progression in GA patients using only OCT volumes in a time-continuous manner. The model could provide an eye-specific risk map, which could reveal the regions more susceptible to developing RORA. The data were collected from 129 eyes belonging to 119 patients with GA who visited the Jules Gonin Eye Hospital in Lausanne, Switzerland. The longitudinal OCT images were acquired with the Heidelberg Spectralis device using a 6 × 6 mm macular cube (49 or more B-scans) with follow-up mode. The dataset was split into a training set with 109 eyes (99 patients) and a test set with 20 eyes (20 patients). The latest definition of RORA was used for the assessment [33]. For ground-truth (GT) training, the RORA regions were automatically segmented from each B-scan in the training set using a DL model. The model used the 0.5 projected presence probability threshold to create a binary en face RORA segmentation. Additionally, the retinal layers and drusen were automatically segmented from all OCTs in the train and test set and transformed to en face thickness maps, which were used as input for the atrophy progression algorithm. The CNN algorithm relied on an encoder–decoder architecture built upon an EfficientNet-b3 model pre-trained on Imagenet. The model receives a 13-channel input (13 en face maps: 6 layer thickness maps with their 6 corresponding reflectance maps + 1 drusen height map) and outputs a K of 5 channels, which can then be used as parameters in a time-based Taylor series for each en face pixel. The time-based series then allows the progression prediction of RORA segmentation at any given time point, visualized as an atrophy progression risk map, which shows the time for RORA conversion for each pixel. From baseline OCT up to 5 years, the time to conversion was noted as the earliest time point where the RORA probability for an en face pixel exceeded 0.5. Two scenarios were considered: A—the baseline visit was used as input for the model with RORA predictions for baseline and future visits; B—the preceding visit was used as input for the model with RORA predictions for the next available visit. By calculating for each pixel the earliest time point where the RORA probability prediction surpassed 0.5, an atrophy time-to-conversion en face risk map was built with a color-coded time scale. The correlation coefficient (CC) for the RORA progression rate between the manual annotations and the model’s prediction was 0.52. The performance of the algorithm was compared to both manual- and automatic-segmentation GT for each scenario. For both, the DIs were calculated considering total RORA and growth-only RORA areas (highlighted in Table 6). The performance was also evaluated using the square root of the RORA area error calculated for the en face view. When considering manually segmented GT, for scenario A, the average for the total RORA DI ranged from 0.73 to 0.80, while for RORA growth area, it ranged from 0.46 to 0.72. The square root area error had mean values ranging from 0.13 mm to 0.33 mm. For scenario B, the DI for total RORA area had values ranging between 0.83 to 0.88, and for RORA growth area, they ranged from 0.44 to 0.64, with the mean square root area error between 0.16 mm and 0.21 mm. When considering automatically segmented GT, for scenario A, the DI for the total RORA area ranged between 0.74 and 0.85, the growth-only RORA area between 0.39 and 0.71 and the mean square root area error between 0.20 mm and 0.35 mm. For scenario B, the DI for the total RORA area ranged between 0.84 and 0.89, and for the growth-only RORA area, it was between 0.35 and 0.62, while the mean square root area error was between 0.17 mm and 0.20 mm.

### 3.9. Prediction of VA Outcome—Classic Machine Learning

Schmidt-Erfurth et al. [29] developed the first model able to predict the VA outcome at 12 months. The model was applied to 614 nAMD eyes from 614 patients participating in the HARBOR trial after they had 12 months of treatment with the anti-VEGF agent Ranibizumab in different doses and regimens. First, SD-OCT volumes underwent motion artifact removal and then were analyzed in a fully automated manner with algorithms based on graph theory combined with CNNs that segmented the retinal layers and the OCT lesions associated with CNV, such as IRF, SRF and PED. The segmentation of the total retinal thickness together with the aforementioned lesions resulted in four morphologic maps that were further spatially localized in nine zones with the ETDRS macular grid. After segmentation, these extracted retinal biomarkers and the BCVA from baseline and months one, two and three, as well as the anti-VEGF dose and treatment regimen, were used to predict the BCVA at 12 months using random forest regression. The model improved as the number of visits used for prediction increased, with the highest prediction accuracy being at 3 months (from baseline to the third month) with an $R^{2}$ = 0.70 and root mean square error (RMSE) of 8.6 letters. The authors also measured the structure–function correlation at baseline and observed that, among the morphological features, IRF horizontal extension in the 1 mm and 3 mm foveal area + volume in the central 1 mm played the most important roles in BCVA, followed by total retinal thickness. When using the baseline visit to predict BCVA at 12 months, the most important feature was BCVA at baseline, followed by IRF area and volume. Interestingly the importance of the BCVA as a feature increased as the number of visits increased, with BCVA at the third month leading the way, while quite the opposite was shown by the morphological features, with their importance starting to decline. The authors pointed out that further studies taking into account the ellipsoid zone and the external limiting membrane using automatic segmentation strategies are needed because of their strong connections with BCVA.

Rohm et al. [31] developed a similar model for predicting BCVA at 3 and 12 months after the initiation phase with three anti-VEGF injections. The prediction relied on five different ML algorithms (AdaBoost.R2, gradient boosting, random forests, extremely randomized trees and Lasso) using 41 features extracted from electronic medical data contained in a data warehouse and 124 spatially resolved measurements obtained from the OCT device’s segmentation software (in the extensive markup language (XML) file associated with the OCT). Besides this, features from different time frames (3 or 12 months) were aggregated along with the measurements, such as mean, variance, minimum and maximum, that were available. The dataset included nAMD patients treated with anti-VEGF at Ludwig Maximilian University, Munich, Germany. The paper included 738 eyes from 653 patients for the 3-month forecast and only 508 eyes from 456 patients for the 12-month forecast because of insufficient long-term data. All of the five models mentioned earlier were evaluated in a 10-fold cross-validation. The models’ performances with this retrospective RWD dataset yielded the following errors between the predicted VA and the ground truth for the 3-month forecast (VA expressed as the logarithm of the minimum angle of resolution (logMAR), with the equivalent in ETDRS letters in parentheses): 0.11 to 0.18 logMAR (5.5 to 9 letters) MAE and 0.14 to 0.20 logMAR (7 to 10 letters) RMSE. For the 12-month forecast, they obtained the following: 0.16 to 0.22 logMAR (8 to 11 letters) MAE and 0.20 to 0.26 logMAR (10 to 13 letters) RMSE. The model’s RMSE increased when adding information from multiple visits, with the lowest RMSE of 0.20 logMAR (10 letters) being obtained when the baseline + visits one, two and three were used, while the MAE did not improve with additional visits. In terms of feature importance, previous BCVA was shown to have a great influence on the models’ predictions. Interestingly, among the five different ML algorithms, the authors obtained the best results using the Lasso protocol.

### 3.10. Prediction of VA Outcome—Deep Learning

Kawczynski et al. [30] developed DL models to predict the BCVA of nAMD patients using data from the phase-three HARBOR clinical trial totaling 1097 patients with treatment-naive subfoveal CNV secondary to nAMD. In the trial, one eye per patient was randomized in a 1:1:1:1 fashion to Ranibizumab with the following scenarios: (1) 0.5 mg monthly, (2) 0.5 mg PRN, (3) 2.0 mg monthly and (4) 2.0 mg PRN. Patients in the PRN groups were given three monthly injections and afterwards treated only if there was a sign of disease activity on the OCT images or if the BCVA dropped ≥ five letters from the previous visit. The BCVA results were measured with ETDRS charts and OCT images were captured with Cirrus HD-OCT (Carl Zeiss Meditec, Dublin, CA, USA) monthly for 2 years. The final dataset containing 1071 patients was split at the patient level into a training set with 924 patients, further divided in five groups for cross-validation, and an evaluation set with 147 randomly selected patients. The models were evaluated based on a ResNet-50 v2 CNN architecture in four scenarios: predicting the exact BCVA in ETDRS letters from OCT images (1) at the current visit and (2) at 12 months from the baseline OCT and predicting the BCVA under a certain threshold (<69 letters (Snellen equivalent, 20/40), <59 letters (Snellen equivalent, 20/60) or ≤38 letters (Snellen equivalent, 20/200)) from OCT images obtained (3) at the current visit and (4) at 12 months from the baseline OCT. A detailed description of the results is shown in Table 7. The prediction of BCVA at the current visit had $R^{2}$ = 0.67 (RMSE = 8.60 letters) for study eyes and $R^{2}$ = 0.84 (RMSE = 9.01 letters) for fellow eyes, with the best model achieving AUCs of 0.92 and 0.98 for study eyes and fellow eyes, respectively. Using the baseline OCT, the model’s BCVA predictions at 12 months had $R^{2}$ = 0.33 (RMSE = 14.16 letters) for study eyes and $R^{2}$ = 0.75 (RMSE = 11.27 letters) for fellow eyes. For the classification of BCVA under certain thresholds, the best model achieved AUCs of 0.84 and 0.96 for study eyes and fellow eyes, respectively. The authors concluded that the BCVA prediction task for the study eyes was more difficult since the inclusion criteria range for the BCVA at inclusion was restricted compared to the higher variability in fellow eyes.

## 4. Discussion

We are living in extraordinary times, with medicine, technology and artificial intelligence evolving at an accelerated pace. The use of artificial intelligence has spread in all areas of our life with an emphasis on personalizing our experiences. We can, therefore, expect to see this shift towards personalizing experiences in an essential domain such as healthcare, where physicians strive to tailor the management of their patients to fit their individual needs. As such, we can see the relevance of conducting a thorough and exhaustive systematic review of this topic at this moment in time with the emergence of cutting-edge research in personalized AI AMD management. When compared to other systematic reviews on similar topics (Table 8), the numbers of records identified and studies finally included in the current review further strengthen its relevance.

### 4.1. Prediction of Treatment Requirement

*Dataset and prediction task:* Treatment requirement models were trained on datasets extracted either from clinical trials or from clinical practice. Table 9 includes details about the number of considered patients and their provenance. Except for Pfau [16], all the other datasets had cohorts in the order of hundreds of patients. The prediction task for the treatment requirement was framed as a regression for the number of injections or as a classification in low/medium/high classes for treatment demand or for the average interval in weeks between treatments. The prediction used data available at the beginning of the treatment, generally the first three visits. The condition for high-demand treatment during one year was ≥16 injections for Bogunovic [14] and Romo Bucheli [15], ≥10 injections for Pfau [16] and a ≤5-week interval between injections for Gallardo [17]. Low-demand treatment meant ≤5 injections per year for Bogunovich [14], Romo Bucheli [15] and Pfau [16] and a ≥10-week interval for Gallardo [17].

*Features and algorithms:* Automatically extracted biomarkers across the nine ETDRS regions were present in all the four studies, being the main features for the first three and an alternative approach for the last one. The techniques used to extract them varied from graph theory segmentation to CNN-based semantic segmentation. Classical machine learning algorithms were applied to the extracted features in the first three studies, while an end-to-end deep learning approach was introduced by Romo Bucheli et al. [15], which worked at the level of OCT volumes reduced to 2D representations after several preprocessing steps. The performance of the prediction was greatly improved in the end-to-end approach, since the learning algorithm had access to the images and not only the extracted features. Integrating images in an end-to-end process means that the model is “free” to use any kind of biomarker, known or unknown, as long as the neural networks are able to capture them. The disadvantages are the black box aspect and the increasing data requirement with the increase in the complexity of the neural network (NN) architecture. The first can be mitigated with the occlusion of sensitivity heatmaps and the second through the reduction of the 3D volumes to 2D representations.

*Evolution aspect:* Even though RNNs were used only by Romo Bucheli et al. [15], the differential features between subsequent visits were also present in the work by Gallardo [17] and Bogunovic [14].

*Quality of treatment decision:* Predictive models for the treatment requirements depend on the quality of the decisions taken by the clinicians based on the followed protocol. Furthermore, the treatment process might also be affected by patient-dependent factors in the case of RWD patients. Out of the studied articles, only the study by Romo Bucheli et al. [15] used a curated dataset. A deep learning-based automated quantification fluid algorithm with an AUC of 0.93 was used to detect questionable non-injection events, and patients who showed more than three disagreements between the clinical decision and the decision based on the automatically detected fluid in the entire 2-year treatment period were excluded.

*Qualitative evaluation/performance:* The most common performance metrics used for evaluation of the treatment requirement prediction were the AUC, accuracy and MAE (see Table 10). The values for the AUC ranged from 0.7 to 0.85, meaning that the task was not trivial. The high-demand class tended to be better predicted. In some cases, the decrease in prediction quality when using only the first visit compared to using the baseline + two visits was not significant.

Challenging factors came from the unknown aspects of the disease, differences in treatment protocols that could result in imperfect training data, imbalanced datasets (the number of patients with moderate demand was significantly higher than the number with low/high demand) and errors in extracting the used features (Table 11 highlights the potential biases for studies predicting the treatment requirements).

*Most important features:* Features extracted from the central area were consistently very important for all the models. The common conclusion in the four studied papers was that SRF, IRF and retinal thickness individual measurements, together with their dynamics, played a very important role in the identification of treatment requirements. There was no consensus regarding the most important visit: in the work by Bogunovic [14], the baseline features were less important, while in the work by Gallardo [17], for the low- and moderate-demand classes, the most important features were the ones extracted at the baseline visit. We need to underline that the tools for the extraction of the features strongly evolved between the publication times for Bogunovic [14] vs. Gallardo [17].

The fact that the low-demand class was better identified than the high-demand class only in the end-to-end approach was related, from the authors’ perspective, to the fact that traditional biomarkers related to retinal fluid might be complemented by automatically derived features for patients with low-demand treatment.

### 4.2. Prediction of Treatment Efficacy

*Dataset and prediction task:* Treatment efficacy models mainly used RWD datasets, with three datasets belonging to hospitals from China and only one dataset belonging to a hospital from South Korea. Table 12 contains details regarding the number of patients included and their provenance. The size of the datasets ranged from 206 to 526 patients. The prediction task for Feng et al. [18] and Zhao et al. [21] was framed as a classification task, while for Liu et al. [20] and Lee et al. [19], it was framed as a generative task evaluated with a classification task. Feng’s model [18] was asked to classify the pre-therapeutic images based on the clinicians’ judgment of efficacy. Liu et al. [20] tested the GAN-generated post-therapeutic image’s capacity for the classification of the macular appearance (wet/dry or if it had transformed from wet to dry). Lee et al. [19] also used a post-therapeutic GAN-generated image and assessed its capacity to predict the presence of macular lesions (IRF, SRF, SHRM and PED). Zhao et al. [21] tested the model’s ability to use pre-therapeutic images in order to classify patients as responders and non-responders based on the change in BCVA. All models used OCT scans and all of them used the pre- and post-therapeutic images, except for Feng et al. [18], who used only the pre-therapeutic images. The input and prediction tasks are displayed in Table 13.

*Features and algorithms:* Since all models employed DL methods, they received 2D OCT scans as input from only one visit. Feng et al. [18] used ResNet-50, Liu et al. [20] used a pix2pixHD GAN, Lee et al. [19] used a cGAN and Zhao et al. [21] used SSG-Net. Feng’s ResNet-50 [18] was tested using either the full OCT image or only the cropped lesion region, and a higher performance was observed when the entire image was used, therefore suggesting the necessity to extend the research for biomarkers beyond the already well-known features. The data augmentation did not necessarily improve the overall performance of the model but created a more balanced dataset, as seen in the increased balance between sensitivity and specificity outlined in Table 14. Relying only on OCT scans for prediction is beneficial when large amounts of data without annotations are available, with experts in the field using the power of unsupervised and self-supervised OCT learning as a method for biomarker discovery [37,38]. However, this approach pushes us away from explainability. Among the four articles included, only Zhao et al.’s [21] used class activation mapping as a method to visualize the regions with more impact on the SSG-Net’s prediction. For differentiating between responders and non-responders, the color-coded heatmap for SSG-Net showed that it took into greater consideration regions associated with visual changes, such as IRF, SRF, PED and even epiretinal membrane, compared to ResNet-50 (in the machine–machine comparison employed by Zhao et al. [21]), which focused its attention more on the macular area and ignored the SRF.

*Various interpretations for treatment efficacy:* The performance of these models in predicting treatment efficacy was closely connected to the quality of the decisions taken by clinicians in considering the anti-VEGF treatment as effective or the post-therapeutic generated image as appropriate. Even though the efficacy was evaluated in different ways, all authors except Feng et al. [18] provided explanations regarding their decisions (Table 15 presents the potential biases for studies predicting the treatment efficacy). The pix2pixHD GAN used the generated post-anti-VEGF therapeutic image to predict the macular status as being either wet or dry and if the conversion from wet-to-dry had occurred. The authors appreciated the prediction of the macular status as being dry (without IRF or SRF) or wet (with IRF or SRF). The cGAN used the generated post-anti-VEGF therapeutic image to predict the presence of IRF, SRF, PED and SHRM 30 days after patients finished their anti-VEGF loading phase. The presence of the lesions in both real and generated post-therapeutic images was graded by two clinicians and two graders. SSG-Net predicted the effectiveness of the anti-VEGF treatment based on the patient’s post-therapeutic BCVA and classified patients as responders (≥1 Snellen line improvement) and non-responders (stabilization or decrease).

*Qualitative evaluation/performance:* As this was considered a classification task, models were evaluated using the AUC, accuracy, sensitivity, specificity, positive predictive value (PPV) and negative predictive value (NPV). It is a complicated task to compare the models’ performances since their prediction tasks and efficiency decisions varied; however, based on accuracy, it seems that, overall, the GAN had higher performances. Since all articles except Lee et al.’s [19] used only OCT images, the addition of FA and ICGA showed improvements in the cGAN’s ability to learn the differences between pre- and post-therapeutic images (higher accuracy for all lesions). Finally, the ResNet-50, pix2pixHD GAN and cGAN models learned the OCT structural outcome after anti-VEGF treatment, while SSG-Net learned the OCT structure–function correlation.
diagnostics-13-02464-t015_Table 15Table 15Potential biases for studies predicting treatment efficacy (green = lower risk for bias, red = higher risk for bias, yellow = unknown/unclear risk for bias; a sample under 100 patients was considered small; RWD = real-world data; RCT = randomized clinical trial; RCLP = real-life clinical practice).AuthorsSmall Sample SizeData SourceExternal ValidationAutomatically Extracted FeaturesQuality of Efficacy InterpretationFeng et al. [18]NoRWDNoN/AUnknownLiu et. al. [20]NoRWDNoN/ARLCP OCT biomarkersLee et al. [19]NoRWDNoN/ARLCP OCT biomarkersZhao et al. [21]NoRWDNoN/ARLCP BCVA

### 4.3. Prediction of Conversion to nAMD

*Dataset and prediction task:* In predicting conversion from non-nAMD and early or iAMD to nAMD, both ML and DL models were employed. These models were trained on datasets extracted either from a clinical trial, such as HARBOR (Schmidt-Erfurth et al. [13] and Banerjee et al. [24]), or from a clinical practice: Moorfields Eye Hospital (Russakoff et al. [22], Yim et al. [23]) or Bascom Palmer Eye Institute (Banerjee et al. [24]). Datasets consisted of non-nAMD or early or iAMD fellow eyes of nAMD patients: 71 (Russakoff et al. [22]), 495 (Schmidt-Erfurth et al. [13], 671 (Banerjee et al. [24]) and 2261 eyes (Yim et al. [23]). A comparative view of the different datasets can be seen in Table 16. The task was to predict the risk of conversion from early or iAMD to nAMD (at 24 months (Russakoff et al. [22]) or (at 3, 6, 9, 12, 15, 18 and 21 months (Banerjee et al. [24])), the risk of conversion from iAMD to nAMD (at 24 months (Schmidt-Erfurth et al. [13] or the risk of conversion from non-nAMD to nAMD (at 6 months (Yim et al. [23])).

*Features and algorithms:* Both ML and DL models were used in predicting the conversion from non-AMD or early or iAMD to nAMD. Schmidt-Erfurth [13] used a sparse CPH model, while the others used DL models: AMDnet and VGG16 were tested by Russakoff [22], Deep Sequence LSTM by Banerjee [24] and an ensemble DL model by Yim [23]. The models were fed with either extracted OCT features or OCT scans, with a few variations in terms of supplementary features and the number of visits (detailed in Table 13). Schmidt-Erfurth et al. [13] added, alongside quantitative spatiotemporal features extracted from OCT images, demographic features, smoking status and genetic data (SNPs). Russakoff et al. [22] used only OCT B-scans. Banerjee et al. [24] used quantitative spatiotemporal features extracted from OCT images, age, gender, race, smoking status and BCVA. Yim’s model [23] was fed with 3D raw OCT images + 3D one-hot-encoded OCT segmentation maps. The numbers of follow-up visits used by the models were different: baseline + four visits for the sparse CPH model; baseline for AMDnet and VGG16; total visits available for 3, 6, 9, 12, 15, 18 and 21 months and baseline + four visits for 18 months for the Deep Sequence model; and one visit for the Ensemble DL model. The one visit used by Yim et al. [23] for the 6-month prediction employed either the conversion scan or the injection scan. A comparative view of the number of visits and the prediction time frame can be seen in Table 17.

*Quality of nAMD conversion decision:* The moment of conversion to nAMD was determined manually by two graders in Schmidt-Erfurth’s study [13] and it was based on the presence of either IRF or SRF with associated suspicious PED or SHRM. In Russakoff’s study [22], conversion was determined as new-onset macular fluid in OCT scans, confirmed by FA, that showed the presence of CNV. Yim et al. [23] used two criteria—(1) the moment the patient started anti-VEGF treatment, named the injection scan; and (2) the moment of conversion, as decided by expert review of OCT images—defining nAMD conversion as the presence of IRF or SRF associated with a suspicious PED, hemorrhage or SHRM. In Banerjee’s study [24], the nAMD conversion in the dataset from the HARBOR trial was confirmed by the Digital Angiography Reading Center using multimodal imaging, and the same approach was used by a clinical reading center for the MIAMI dataset (Table 18 displays the potential biases for studies predicting conversion to nAMD).

*Qualitative evaluation/performance:* When evaluating the risk of conversion from non-nAMD or early or iAMD to nAMD, all the models used the AUC as the performance metric. AUC values were somewhat similar, with greater values for Russakoff’s model [22] (0.91 for 24-month prediction) and Banerjee’s model [24] (0.96 for 3 months, 0.97 for 21 months), but only 0.68 for 24 months with Schmidt-Erfurth et al.’s model [13]. Model performance significantly improved if pre-processing steps were involved. For Banerjee’s Deep Sequence model [24], a drop in performance was noticed for 12 months (0.77 ± 0.06), but performance increased for 18 months (0.9 ± 0.06) and 21 months (0.97 ± 0.02), presumably pinned to the fewer sequential follow-up visits available for training for these time frames and the more similar characteristics of converting and non-converting classes. The Deep Sequence model generalized well when tested on the external Bascom Palmer Eye Institute dataset for shorter time intervals, with 0.82 for 3 months, but wide distributional shifts between the two datasets did not favor the prediction for longer time intervals, with 0.77 for 6 months, 0.69 for 12 months and 0.68 for 21 months. Yim et al.’s model [23] reached an AUC of 0.75 on the test set for the volumetric OCT conversion scan and 0.745 on the test for volumetric OCT injection scan.

*Most important features:* For Schmidt-Erfurth’s model [13], the increase in drusen area, drusen-centric HRF and thickening of the ONL in regions of HRF concentration were the most important features. Russakoff’s model [22] highlighted that pixels under the RPE or involving the choroid seemed to be important as opposed to the pixels around the RPE for those who were not progressing. Banerjee’s model [24] analyzed the 21 imaging features related to the presence of drusen and their number, extent, density and relative reflectivity. Yim et al.’s model [23] sensitivity improved when the extracted features related to HRF, high drusen volumes and PED were present.

### 4.4. Prediction of Conversion to GA

*Dataset and prediction task:* The conversion to GA models used both RWD (Rivail et al. [25] and data from clinical trials (HARBOR clinical trial—Schmidt-Erfurth et al. [13]). Even though the number of patients was different between the two studies, the number of eyes with iAMD included was similar (420 vs. 495), as shown in Table 19. Patients in the HARBOR clinical trial had monthly OCT follow-ups (HD-OCT device—Cirrus; Carl Zeiss Meditec, Inc., Dublin, CA, USA) for up to 2 years, while the patients from Rivail’s dataset [25] had OCT follow-ups (Spectralis OCT device—Heidelberg Engineering, Heidelberg, Germany) every 3 or 6 months for up to 7 years. Schmidt-Erfurth et al. [13] followed the conversion to both nAMD and GA in 495 eyes: 114 converted to nAMD, 45 eyes converted to GA and 336 iAMD eyes remained stable during the 2 year study. In Rivail’s study [25], from 420 iAMD eyes, 48 eyes converted to GA; therefore, it was similar to Schmidt-Erfurth’s study [13]. Schmidt-Erfurth predicted the conversion to GA in a 24-month time frame using the baseline visit + the first four additional follow-up visits, while Rivail [25] predicted for three different time intervals, 6, 12 and 18 months, using only one visit, as seen in Table 20.

*Features and algorithms:* Both ML and DL models were used, with Schmidt-Erfurth using a sparse CPH model [13] and Rivail a deep Siamese network [25]. The sparse CPH model received quantitative spatiotemporal features extracted from the OCT (at baseline and over the following 4 months) with additional demographic features, smoking status and AMD genetic information (SNPs), while the deep Siamese network received only OCT B-scans (shown in Table 13). The deep Siamese network was first pre-trained on a pretext task to estimate the time interval between pairs of images from the same patient, essentially creating an aging model. Compared to others, this type of learning thrives with large, unlabeled longitudinal datasets where the intervals between visits are not regular, the registration is not ideal or there is not sufficient time or manpower for annotating the data. Learning this pretext task improved the model’s performance in predicting the conversion to GA. On the other hand, the Cox proportional hazard model was trained on the HARBOR clinical trial dataset and thus benefited from regular sampling intervals and quantitative features automatically extracted from the OCT.

*Quality of GA conversion decision:* In Schmidt-Erfurth’s study [13], using OCT imaging, two graders manually determined the time to conversion to both nAMD and GA, while in Rivail’s study [25], the method for identifying patients who converted to GA was not specified (Table 21 shows the potential biases for studies predicting the conversion to GA). Considering that patients within the HARBOR clinical trial had monthly OCT follow-ups, this could provide a more accurate representation of the actual moment of conversion versus the 3- or 6-month OCT follow-ups in Rivail’s study [25], which could have missed the moment of conversion by a few months. However, Rivail [25] followed patients for a longer period of time, up to 7 years, thus having a higher chance to detect those patients who had later conversion.

*Qualitative evaluation/performance:* The prediction task for correctly identifying the eyes that would convert to GA in a given time-frame can be framed as a classification task that sorts iAMD patients into converters and non-converters; therefore, it was evaluated in terms of AUC, sensitivity, specificity and precision. While there were great differences in the two approaches, with different models, inputs and numbers of visits, the results measured as the AUC were quite similar for the longer time intervals, with a slightly higher performance for the CPH (AUC: 0.80 (for 24 months) vs. 0.773 (for 18 months)) with the deep Siamese network (displayed in Table 20). The sparse CPH prioritized sensitivity over specificity, and this seems reasonable since it is important to identify as many converting eyes as possible, even though there is a chance of mislabeling a few non-converters as converters. Sensitivity and specificity were not discussed in Rivail’s study [25]; however, the precision metric was available and the highest performance was for the 18-month interval with a precision 0.463, meaning that, among converters, there were many falsely labeled as such. Additional research involving DL models with more follow-up visits used for learning and monthly visits during the follow-up scan could allow a more clear comparison versus classical ML models.

*Most important features:* Schmidt-Erfurth’s study [13] demonstrated that iAMD eyes that convert to nAMD or GA follow distinct pathways in terms of feature evolution. For GA conversion, the following features ranked among the most important: atrophy at the level of the RPE + IS/OS segment with irregularity, reduced thickness and volume, increased HRF in the ONL layer, thinning of the ONL and age. Additional research regarding the importance of features in DL models predicting conversion to GA through different explainability methods is necessary in order to obtain a better understanding of the decision-making process of more complex models and to identify whether OCT imaging data other than the commonly extracted quantitative features also play a role in predicting iAMD patients at risk for GA.

### 4.5. Prediction of GA Growth

*Dataset and prediction task:* All GA growth models used RWD datasets collected from Switzerland, the United States and China. The number of patients included ranged from 25 to 119, with different numbers of visits and visit intervals (presented in Table 22); numbers were smaller compared to the datasets used for the other types of predictions. All models sought to predict the growth of GA lesions at future time points (Gigon et al. [28] described GA using the most recent definition of RORA [33]). The prediction task should be seen as a pixel classification task, where future GA maps are generated and used to discern the regions that will develop GA. All models were tested in various scenarios with different numbers of training visits and different time points for prediction (differences are highlighted in Table 23). Gigon et al. [28] used the baseline visit in one scenario and the preceding visit in a different scenario, Niu et al. [26] used the baseline + first follow-up visit and Zhang et al. [27] used the baseline + first two sequential follow-up visits.

*Features and algorithms:* In predicting GA growth, both ML and DL models were employed. Niu et al. [26] used a random forest classifier, Zhang et al. [27] used BiLSTM with a 3D-UNet CNN refinement and Gigon et al. [28] used an encoder–decoder CNN with EfficientNet-b3 as the backbone + time-based Taylor series. The random forest classifier predicted at a given time whether a pixel in the topographic image would develop GA or not, the BiLSTM with a 3D-UNET CNN had as output a GA projection color-coded map for future GA growth regions and the encoder–decoder CNN had as output en face time-to-RORA-conversion risk maps. The random forest classifier received 19 quantitative spatiotemporal features extracted from the OCT images at baseline and the first visit, the BiLSTM + 3D-UNet was fed OCT volumes + time factors from the baseline and first two sequential follow-up visits and the encoder–decoder CNN used 13 en face segmentation maps from baseline after automatically segmenting retinal layers and drusen. The advantage of Gigon’s approach [28] is that it allows for the prediction of RORA progression at flexible time intervals, providing the ability to track future changes in a time-continuous manner. The features used by the random forest classifier described the GA lesion, drusen, pseudodrusen, retinal intensity, photoreceptor loss, axial thickness and reflectivity between retinal boundaries. Six-layer thickness + reflectance maps (retinal nerve fiber layer, ganglion cell layer + internal plexiform layer, ONL, photoreceptors + RPE, choriocapillaris + Sattler’s layer) and drusen height were used by the encoder–decoder CNN. We can see that both the random forest model and the encoder–decoder CNN took into account drusen, retinal layer thicknesses and reflectivity measurements.

*Quality of ground truth/performance:* The performance of the GA growth models depended on the precision of the GT used for comparison. Niu [26] and Zhang [27] used automatic segmentation for the GT and visually reviewed it, while Gigon [28] used both manual and automatic segmentation techniques. Gigon et al. [28] tested their model on both manually and automatically segmented GT and concluded that the average DI and area errors were very similar between the two; therefore, automatically segmented GT came close to the expert’s segmentation (Table 24 highlights the potential biases associated with studies predicting GA growth).

*Qualitative evaluation/performance:* The common metric used for performance evaluation was the DI and the results are displayed in Table 23. By comparing the different scenarios, one can see that models were more powerful when given more follow-up visits and integrating time factors, while more follow-up visits from the same patient increased their performance even further. The highest AUC of 0.92 was obtained when the BiLSTM + 3D-UNet used all prior consecutive follow-ups to predict the last follow-up visit for each patient. This highlights that similar patient-specific prior information is more helpful in predicting GA growth in spite of the many similarities in GA growth between patients.

*Most important features:* Since the random forest classifier relied on extracted OCT features, feature order of importance was calculated and the top candidates in predicting the location of GA growth were the regions with photoreceptor loss, lower reflectivity for the ellipsoid zone and decreased thickness for reticular pseudodrusen.

### 4.6. Prediction of VA Outcome

*Datasets and prediction task*: The datasets used to predict the VA outcome contained between 614 and 1246 nAMD eyes, as displayed in Table 25. The prediction tasks targeted two time horizons: either 3 months or 12 months.

*Features and algorithms*: Classical ML was based on ensemble methods (e.g., random forest regression, gradient boosting, AdaBoost, extremely randomized trees) or regression analysis (i.e., lasso). Deep learning was based on the ResNet-50 v2 architecture. While classical ML relied on four visits (baseline plus three more visits) to predict VA outcome for both 3 and 12 months, deep learning was more ambitious, aiming to predict VA outcome at 12 months using only the first visit (differences highlighted in Table 26).

*VA measurement quality:* The performance of the VA outcome models was highly dependent on the quality of the VA measurements. Both Schmidt-Erfurth [29] and Kawczynski [30] used the HARBOR dataset for their models, where BCVA was measured in clinical trial settings using standard ETDRS charts, therefore offering more reliability in terms of measurement accuracy. Rohm et al. [31] used RWD BCVA measurements uploaded to the data warehouse from patients visiting Ludwig Maximilian University of Munich, which can be more prone to error than standardized clinical trial VA measurement settings (Table 27 presents the potential biases associated with studies predicting the VA outcome). Even though RCT-derived BCVA offers more credibility, errors might still appear, as shown by the variability in the intersession measurements in AMD patients [39].
diagnostics-13-02464-t025_Table 25Table 25Machine learning and deep learning models for predicting VA outcome and the datasets used for prediction.AuthorsDateModelDataset**Machine Learning**Schmidt-Erfurth et al. [29]January 2018Random forest regression614 nAMD eyes of 614 patients undergoing anti-VEGF treatment within the HARBOR clinical trialRohm et al. [31]July 2018AdaBoost.R2, gradient boosting, random forests, extremely randomized trees, lasso738 nAMD eyes from 653 patients (for 3-month forecast) and 508 nAMD eyes from 456 patients (for 12-month forecast) visiting Ludwig Maximilian University of Munich, Germany**Deep Learning**Kawczynski et al. [30]September 2020ResNet-50 v21071 nAMD eyes of 1071 patients undergoing anti-VEGF treatment within the HARBOR clinical trial
diagnostics-13-02464-t026_Table 26Table 26Machine learning and deep learning models for predicting VA outcome (RMSE = root mean squared error; BL = baseline; V = visits; BCVA = best-corrected visual acuity; m = months).Author and PredictionModel and ScenarioNo. of VisitsPrediction TimeR^2^RMSE**Machine Learning**Schmidt-Erfurth et al. [29]—BCVA at 12 monthsRandom forest regressionBL + 3 V12 m0.708.6 lettersRohm et al. [31]—BCVA at 3 and 12 monthsAdaBoost.R2BL + 3 V12 mN/A0.2 logMARAdaBoost.R2BL + 3 V3 mN/A0.25 logMARGradient boostingBL + 3 V12 mN/A0.19 logMARGradient boostingBL + 3 V3 mN/A0.2 logMARRandom forestsBL + 3 V12 mN/A0.18 logMARRandom forestsBL + 3 V3 mN/A0.23 logMARExtremely randomized treesBL + 3 V12 mN/A0.18 logMARExtremely randomized treesBL + 3 V3 mN/A0.23 logMARLassoBL + 3 V12 mN/A0.2 logMARLassoBL + 3 V3 mN/A0.18 logMAR**Deep Learning**Kawczynski et al. [30].—BCVA at current visit and 12 monthsResNet-50 v2—study eyesBL12 m0.3314.16 lettersResNet-50 v2—fellow eyesBL12 m0.7511.27 letters
diagnostics-13-02464-t027_Table 27Table 27Potential biases for studies predicting the VA outcome (green = lower risk for bias, red = higher risk for bias; a sample under 100 patients was considered small; RWD = real-world data; RCT = randomized clinical trial; RCLP = real-life clinical practice; N/A = not applicable).AuthorsSmall Sample SizeData SourceExternal ValidationAutomatically Extracted FeaturesQuality of VA MeasurementSchmidt-Erfurth et al. [29]NoRCTNoYesRCT measurementRohm et al. [31]NoRWDNoYesRCLP measurementKawczynski et al. [30]NoRCTNoN/ARCT measurement


*Qualitative evaluation/performance*: For the task of predicting BCVA at 12 months only from baseline OCT and BCVA, the model developed by Schmidt-Erfurth [29], which used handcrafted OCT features with a subset of 614 patients from the HARBOR dataset, reported $R^{2}$ = 0.34. For the same task, Kawczynski’s model [30] achieved $R^{2}$ = 0.45 with the 924-patient training set and $R^{2}$ = 0.40 for the 126-patient test set. We can therefore presume that the greater variability explained by the DL models might have come from considering the whole image and taking into account features still unknown among handcrafted feature engineers. Both studies had higher results when both the OCT and the BCVA were used in their predictions. Besides the $R^{2}$ value, the root mean squared error (RMSE) was also used, with the best obtained value being 8.5 letters or 0.18 on the logarithm of the minimum angle of resolution (logMAR) chart. These best results were obtained with extremely randomized trees and lasso for a 12-month prediction.

### 4.7. Quality Assurance for AI Models

Efforts have been made in order to assess the risk of bias in AI models. The International Telecommunication Union (ITU) partnered with the World Health Organization (WHO) to create a Focus Group on Artificial Intelligence for Health (FG-AI4H) [40] in order to develop a standardized assessment framework for health-based AI models. As such, the proposed AI audit, carried out at different stages of a model’s deployment, represents a detailed standardized report that can reveal a model’s strengths and limitations [41]. A detailed audit report for diabetic retinopathy screening using the robust FG-AI4H template was presented by Oala et al. [42]. These type of interventions involving auditing and quality control set the path towards efficient and safe implementation of AI models in real-world clinical settings.

### 4.8. Challenges for AI in Real-World Clinical Settings

*Data shortages*. For upcoming DL models to thrive, we are still in need of larger, more diverse, publicly accessible OCT datasets for AMD patients with multiple follow-up visits. In overcoming this shortage by making use of smaller datasets, various techniques can be implemented, such as using OCT neighboring B-scans, transfer learning and data augmentation [43]. Khan et al. addressed this limitation by reviewing publicly available datasets for ophthalmology imaging [44]. Along the same lines, researchers interested in finding OCT datasets might have an eye on the ongoing Common European Data Spaces, one of which is dedicated to health. Meanwhile, the datasearch service from Google has become a valuable tool for quick identification of public datasets.

*OCT devices and acquisition protocol*. The acquisition protocol varies between different OCT devices but also within the same device, and this represents a challenge for AI-assisted volumetric analysis of OCT images, which would be favored by a standardized approach. In this context, methodologies like “data FAIRification” are valuable tools to assess the quality of a dataset.

*Data reliability*. Data reliability has yet another dimension. The performance metrics were reported under the assumption that the ground truth used for training was 100% accurate. The extent to which this assumption is wrong was signaled by Cabitza et al. [45]. For instance, in diagnostics, the average accuracy of medical experts ranges from 80% to 90%, while the average error rate among radiologists is around 30%. Cabitza et al. computed the number of annotators required to achieve a 95% accurate ground truth. For an error rate of 20%, one needs 10 raters, while for an error rate of 30%, the number of raters should be no less than 25. It is simply not feasible to have this number of raters for the same image. Cabitza et al. name this problem “the elephant in the machine”. To handle it, better metrics are needed to evaluate performance by also considering the interagreement score or the confidence and expertise of the annotator.

*Real-world validation*. Banerjee et al. [24] put in perspective the implementation of prediction algorithms in clinical practice. There are still a lot of unanswered questions regarding whether they can help initiate earlier treatments for exudative events or unburden practices by selecting the patients at higher risk who need monitoring. Another issue, as shown by the current review, is that the majority of studies lack external validation. In order to be able to answer these questions, draw meaningful conclusions and introduce AI prediction algorithms in real-world settings, we are in need of clinical trials to test these outcomes.

*Black-box phenomenon*. The topic of the black-box phenomenon, also taken into account by Schmidt-Erfurth et al. [6], is of utmost importance. While artificial neural networks have impressive results, even outperforming their human pairs, developers and physicians have a hard time understanding the rationale behind these decisions. Interestingly, minor changes [46,47] not visible to the human eye can profoundly alter the classification task performed by a deep neural network and, reversely, images with no meaning for humans can be correctly classified, as shown by Nguyen et al. [48]. The lack of understanding could be problematic in a setting where neural networks could provoke harm through erroneous outputs, leaving the end users blindfolded without the ability to prevent upcoming errors. There seems to be a conflict between performance and explainability, with the model with higher accuracy (e.g., DL) having the least explainable actions and the one with lower accuracy having the opposite [49]. Thus, we can see the value brought by explainable AI (xAI), which has been assigned the mission of providing explanations of a model’s actions to make it more comprehensible to human users, thus widening the clinical acceptance and adoption of AI systems. In order to “break” the black box, there are a number of visualization techniques that can be used.

## 5. Conclusions

A great amount of recent research supports the effectiveness of AI in predicting the progression of AMD during the natural evolution of the disease or after treatment. AI models could substantially improve clinical practice in term of patient selection, treatment selection, developing drugs and establishing valuable trial endpoints. Implementation of prediction algorithms in clinical practice would be of great value for both patients and ophthalmologists. However, this can be perceived as an ambitious task due to the limitations of the analyzed studies and the challenges faced by AI algorithms in real-world clinical settings. As such, there are still a lot of unanswered questions regarding the predictive value of AI models for the future progression of AMD, opening up new avenues for future research.

## Figures and Tables

**Figure 1 diagnostics-13-02464-f001:**
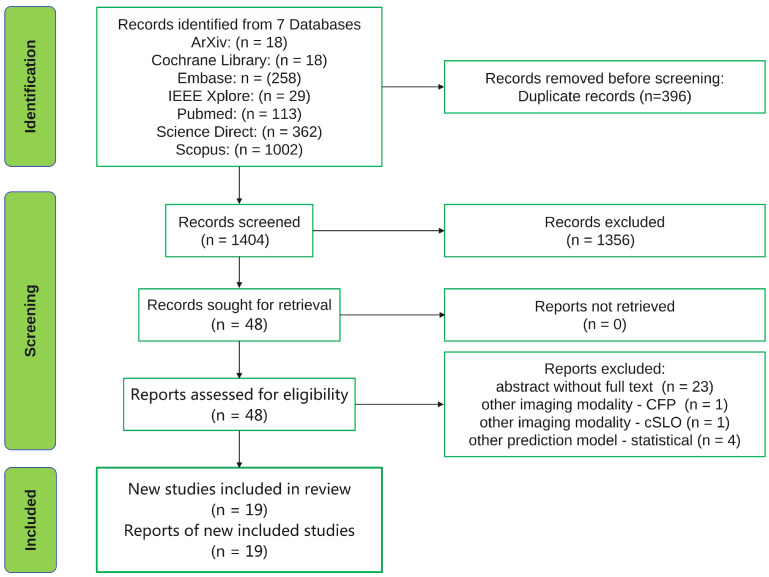
PRISMA flow chart.

**Table 3 diagnostics-13-02464-t003:** The performance of the modified ResNet-50 developed by Feng et al. [18] in predicting anti-VEGF efficiency with four different datasets (Full w DA = full OCT image with data augmentation; Full w/o DA = full OCT image without data augmentation; Lesion w DA = only the region of interest with data augmentation; Lesion w/o DA = only the region of interest without data augmentation) (AUC = area under the curve; DA = data augmentation; w = with; w/o = without).

	AUC	Accuracy	Sensitivity	Specificity
**Full w DA**	0.81	0.72	0.78	0.71
**Full w/o DA**	0.84	0.77	0.98	0.18
**Lesion w DA**	0.83	0.75	0.82	0.63
**Lesion w/o DA**	0.73	0.74	0.96	0.13

**Table 4 diagnostics-13-02464-t004:** The three different scenarios tested by Niu et al. [26] and performances predicting total and growth-only GA regions. (DI = Dice index; SD = standard deviation).

GA Area	Scenario One	Scenario Two	Scenario Three
	**Mean DI +/− SD**		
**Total GA**	0.81 +/− 0.12	0.84 +/− 0.10	0.87 +/− 0.06
**Growth-only GA**	0.72 +/− 0.18	0.74 +/− 0.17	0.72 +/− 0.22

**Table 5 diagnostics-13-02464-t005:** The six different scenarios for GA growth prediction tested by Zhang et al. [27] and their performances (DI = Dice index; CC = correlation coefficient).

Scenario	Predicted GA Growth	Training Data	DI	CC
1	All consecutive follow-up visits for each patient	First two follow-up visits from all other patients	0.86	0.83
2	All consecutive follow-up visits for each patient	All consecutive follow-up visits from all other patients	0.89	0.84
3	Third and subsequent follow-up visits for each patient	First two follow-up visits from the same patient	0.89	0.86
4	Last follow-up visit for each patient	All prior, consecutive follow-up visits from the same patient	0.92	0.88
5	Third and subsequent follow-up visits for all patient	First two follow-up visits from all patients	0.88	0.85
6	Last follow-up visit for all patients	All prior, consecutive follow-up visits from all patients	0.90	0.86

**Table 6 diagnostics-13-02464-t006:** The two different scenarios for RORA growth prediction tested by Gigon et al. [28] and their performances as compared to manually and automatically segmented GT (RORA = retinal pigment epithelium and outer retina atrophy; DI = Dice index; GT = ground truth).

RORA Area	Manually Segmented GT		Automatically Segmented GT	
	**Mean DI**		**Mean DI**	
	Scenario A	Scenario B	Scenario A	Scenario B
Total RORA	0.73 to 0.80	0.83 to 0.88	0.74 to 0.81	0.84 to 0.89
Growth-only RORA	0.46 to 0.72	0.44 to 0.64	0.39 to 0.71	0.35 to 0.62

**Table 7 diagnostics-13-02464-t007:** The performance of the ResNet-50 v2 CNN developed by Kawczynski et al. [30] in predicting BCVA both at the CCV visit and at 12 months and under a certain threshold at the CCV and at 12 months (BCVA = best corrected visual acuity; CCV = concurrent visit; $R^{2}$ = R squared; RMSE = root mean squared error; AUC = area under the curve).

	Predicted	<69 Letters	<59 Letters	<38 Letters
**BCVA at CCV**				
Study eyes	$R^{2}$ = 0.67 (RMSE = 8.60)	AUC = 0.89	AUC = 0.92	AUC = 0.92
Fellow eyes	$R^{2}$ = 0.84 (RMSE = 9.01)	AUC = 0.93	AUC = 0.97	AUC = 0.98
**BCVA at 12 m**				
Study eyes	$R^{2}$ = 0.33 (RMSE = 14.16)	AUC = 0.80	AUC = 0.84	AUC = 0.77
Fellow eyes	$R^{2}$ = 0.75 (RMSE = 11.27)	AUC = 0.92	AUC = 0.93	AUC = 0.96

**Table 8 diagnostics-13-02464-t008:** The number of records identified and studies included in the current systematic review and similar reviews (* = records identified after duplicate removal; ** = records identified after duplicate removal, records marked as ineligible by automation tools and records removed for other reasons).

Article	Records Identified	Studies Included
The current systematic review	1404 *	19
Artificial intelligence for the detection of age-related macular degeneration in color fundus photographs: A systematic review and meta-analysis [34]	632 *	19
Diagnostic accuracy of current machine learning classifiers for age-related macular degeneration: A systematic review and meta-analysis [35]	423 *	14
Deep learning for detection of age-related macular degeneration: A systematic review and meta-analysis of diagnostic test accuracy studies [36]	359 **	18

**Table 9 diagnostics-13-02464-t009:** Machine learning and deep learning models for treatment requirements and their datasets.

Authors	Date	Model	Dataset
**Machine Learning**			
Bogunovic et al. [14]	June 2017	Random forest classifier	317 eyes from 317 nAMD patients undergoing PRN treatment within the HARBOR clinical trial
Pfau et al. [16]	June 2021	NGBoost	40 eyes from 37 nAMD patients visiting the Department of Ophthalmology and Visual Sciences, University of Illinois, Chicago, and 59 nAMD eyes from 59 patients visiting the University Eye Hospital Bonn, University of Bonn, Germany
Gallardo et al. [17]	July 2021	Random forest classifier	377 eyes from 340 nAMD patients undergoing T&E treatment at the University Hospital of Bern
**Deep Learning**			
Romo Bucheli et al. [15]	June 2020	DenseNet + RNN trained end to end	350 nAMD patients undergoing PRN treatment

**Table 10 diagnostics-13-02464-t010:** Machine learning and deep learning models for treatment requirements with their prediction tasks, scenarios and results (Acc. = accuracy; inj. = injections; RCT = randomized clinical trial; RWD = real-world data; PRN = pro re nata; T&E = treat and extend; Tx = treatment; BL = baseline; V = visits; N/A = not available).

Author and Prediction Task	Time Frame	Anti-VEGF Protocol	Model and Scenario	Number of Visits	AUC	Acc.
**Machine Learning**						
Bogunovic et al. [14]—low and high anti-VEGF inj. requirements in RCT (no. of inj.)	2 years	PRN	Low-Tx (≤5 inj.) group vs. others	BL + 2 V	0.7	N/A
			High-Tx group (≥16 inj.) vs. others	BL + 2 V	0.77	N/A
Pfau et al. [16]—low and high anti-VEGF inj. requirements in RWD (no. of inj.)	1 year	PRN and T&E	Low-Tx group (≤4 inj.)	BL	0.68	N/A
			High-Tx group (≥10 inj.)	BL	0.69	N/A
Gallardo et al. [17]—low and high anti-VEGF inj. requirements in RWD (interval between inj.)	1 year	T&E	Low-Tx group (≥10 weeks) vs. others	BL + 2 V	0.79	N/A
			High-Tx group (≤5) vs. others	BL + 2 V	0.79	N/A
**Deep Learning**						
Romo Bucheli et al. [15]—Low, medium and high anti-VEGF inj. requirements (no. of inj.)	2 year	PRN	Low-Tx group (≤5)	BL + 2 V	N/A	0.72
			Intermediate-Tx group (>5 and <16)	BL + 2 V	N/A	0.65
			High-Tx group (≥ 16)	BL + 2 V	N/A	0.9
			Low-Tx group (≤5) vs. others	BL + 2 V	0.85	N/A
			High-Tx group (≥16) vs. others	BL + 2 V	0.81	N/A

**Table 11 diagnostics-13-02464-t011:** Potential biases for studies predicting treatment requirements (green = lower risk for bias, red = higher risk for bias, yellow = unknown/unclear risk for bias; a sample under 100 patients was considered small; * = dataset was curated for disagreements regarding treatment decision; RWD = real-world data; RCT = randomized clinical trial; RCLP = real-life clinical practice).

Authors	Small Sample Size	Data Source	External Validation	Automatically Extracted Features	Quality of Treatment Decision
Bogunovic et al. [14]	No	RCT	No	Yes	RCT protocol
Pfau et al. [16]	Yes	RWD	No	Yes	RCLP protocol
Gallardo et al. [17]	No	RWD	No	Yes	RCLP protocol
Romo Bucheli et al. [15]	No	Unknown	No	Yes	RCLP protocol *

**Table 12 diagnostics-13-02464-t012:** Deep learning models for predicting nAMD anti-VEGF treatment efficacy and their datasets (CNV = choroidal neovascularization; CME = cystoid macular edema).

Authors	Date	Model	Dataset
Feng et al. [18]	August 2020	ResNet-50	228 patients with CNV, CME or both treated with anti-VEGF at Second Affiliated Hospital of Xi’an Jiaotong University between 2017 and 2019
Liu et al. [20]	March 2020	pix2pixHD GAN	526 OCT images from nAMD patients treated with anti-VEGF at Peking Union Medical College Hospital between 2018 and 2019
Lee et al. [19]	March 2021	cGAN	309 eyes of 298 nAMD patients treated with anti-VEGF at Konkuk University Medical Center between 2010 and 2019
Zhao et al. [21]	June 2021	SSG-Net	206 eyes of 181 nAMD patients treated with anti-VEGF at Peking Union Medical College Hospital between 2018 and 2020

**Table 13 diagnostics-13-02464-t013:** The inputs and learning tasks for the considered ML and DL models.

Study	Input	Learning Task
Machine learning—Prediction of treatment requirements		
Bogunovic et al. [14]	Quantitative spatiotemporal features extracted from OCT images (at baseline and months one and two) + BCVA + demographic features + FA pattern type + smoking status	Low and high anti-VEGF injection requirements during 2-year PRN schedule in RCT
Gallardo et al. [17]	Quantitative spatiotemporal features extracted from OCT (at baseline and months one and two) + demographic features	Low and high anti-VEGF injection requirements during 1-year T&E schedule in RWCP
Pfau et al. [16]	Quantitative spatiotemporal features extracted from OCT at baseline	Treatment frequency during 1 year and low and high anti-VEGF injection requirements during 1-year PRN and T&E schedule in RWCP
Deep learning—Prediction of treatment requirements		
Romo Bucheli et al. [15]	OCT volumes (at baseline and months one and two)	Low, medium and high anti-VEGF injection requirements during 2-year PRN schedule
Deep learning—Prediction of treatment efficacy		
Feng et al. [18]	Two-dimensional (2D) pre-therapeutic OCT image (full OCT image/lesion region)	Efficacy of anti-VEGF injection for CNV, CME or both at 21 days (3 weeks)
Liu et al. [20]	Two-dimensional (2D) pre- and post-therapeutic OCT image	Post-therapeutic generated image’s capacity to predict the macular status as either wet or dry and macular wet-to-dry conversion after anti-VEGF
Lee et al. [19]	Two-dimensional (2D) pre- and post-therapeutic OCT image and FA and ICGA image	Post-therapeutic generated image’s capacity to predict presence of IRF, SRF, PED and SHRM 30 days after anti-VEGF loading phase
Zhao et al. [21]	Two-dimensional (2D) pre- and post-therapeutic OCT image	Efficacy of anti-VEGF injection for post-treatment BCVA at approx. 4 weeks
Machine learning—Prediction of conversion to late exudative AMD		
Schmidt-Erfurth et al. [13]	Quantitative spatiotemporal features extracted from OCT (at baseline and months one, two, three and four) + demographic features + smoking status + genetic features (SNPs)	Risk for conversion from iAMD to nAMD and GA in the following 2 years
Deep learning—Prediction of conversion to late exudative AMD		
Russakoff et al. [22]	OCT B-scans	Risk for conversion from early or iAMD to nAMD in the following 2 years
Banerjee et al. [24]	Quantitative spatiotemporal features extracted from OCT + age + gender + race + smoking status + VA	Risk for conversion from early or iAMD to nAMD in the following 3, 6, 9, 12, 15, 18 and 21 months
	Quantitative spatiotemporal features extracted from OCT + age + gender + VA	Risk for conversion from early or iAMD to nAMD in the following 3, 6, 9, 12, 15, 18 and 21 months
Yim et al. [23]	Three-dimensional (3D) raw OCT image + 3D one-hot-encoded OCT segmentation map	Risk for conversion from non-nAMD to nAMD in the following 6 months
Machine learning—Prediction of conversion to GA		
Schmidt-Erfurth et al. [13]	Quantitative spatiotemporal features extracted from OCT (at baseline and months one, two, three and four) + demographic features + smoking status + genetic (SNPs)	Risk for conversion from iAMD to nAMD and GA in the following 2 years
Deep learning—Prediction of conversion to GA		
Rivail et al. [25]	OCT B-scans	Risk for conversion from iAMD to GA in the following 6, 12 and 18 months
Machine learning—Prediction of GA growth		
Niu et al. [26]	Quantitative spatiotemporal features extracted from OCT at baseline and first follow-up visit in three different scenarios	Future GA growth regions in three different scenarios (see text for details)
Deep learning—Prediction of GA growth		
Zhang et al. [27]	OCT volumes + time factors at baseline and first two sequential follow-up visits	Future GA growth regions in six different scenarios
Gigon et al. [28]	Thirteen en-face maps: six layer-thickness maps with their six corresponding reflectance maps + one drusen height map (scenario A: from baseline OCT; scenario B: from preceding OCT)	Future RORA growth regions (scenario A: from baseline to future visits; scenario B: from preceding to next visit)
Machine learning—Prediction of VA outcome		
Schmidt-Erfurth et al. [29]	Quantitative spatiotemporal features extracted from OCT (at baseline and months one, two and three) + BCVA + anti-VEGF dose and regimen	BCVA at 12 months
Rohm et al. [31]	Quantitative spatiotemporal features extracted from OCT XML file (at baseline and months one, two and three) + BCVA + 40 clinical features from EMR	BCVA at 3 and 12 months
Deep learning—Prediction of VA outcome		
Kawczynski et al. [30]	SD-OCT volumes	BCVA at current visit and at 12 months; BCVA < threshold (69, 59, 38 letters) at current visit and at 12 months

**Table 14 diagnostics-13-02464-t014:** Deep learning models’ prediction tasks, scenarios and performances for nAMD anti-VEGF treatment efficacy (AUC = area under the curve; Acc. = accuracy; Se. = sensitivity; Sp. = specificity; PPV = positive predictive value; NPV = negative predictive value; CNV = choroidal neovascularization; CME = cystoid macular edema; Full w DA = full OCT image with data augmentation; Full w/o DA = full OCT image without data augmentation; Lesion w DA = only the region of interest with data augmentation; Lesion w/o DA = only the region of interest without data augmentation; IRF = intraretinal fluid; SRF = subretinal fluid; PED = pigment epithelium detachment; SHRM = sub-retinal hyperreflective material).

Authors and Prediction Task	Scenarios	AUC	Acc.	Se.	Sp.	PPV	NPV
Feng et al. [18]—efficacy of anti-VEGF injection for CNV, CME or both at 21 days (3 weeks)	Full w DA	0.81	0.72	0.78	0.71	N/A	N/A
	Full w/o DA	0.84	0.77	0.98	0.18	N/A	N/A
	Lesion w DA	0.83	0.75	0.82	0.63	N/A	N/A
	Lesion w/o DA	0.73	0.74	0.96	0.13	N/A	N/A
Liu et al. [20]—post-therapeutically generated image’s capacity to predict the macular status as either wet or dry and macular wet-to-dry conversion after anti-VEGF	Macular status (wet/dry)	N/A	0.85	0.84	0.86	0.88	0.82
	Wet-to-dry conversion	N/A	0.81	0.83	0.79	0.75	0.86
Lee et al. [19]—post-therapeutically generated image’s capacity to predict the presence of IRF, SRF, PED and SHRM 30 days after anti-VEGF loading phase	IRF (OCT only) IRF (OCT + FA + ICGA)	N/A	89.6	33.3	95.1	40.0	93.6
		N/A	92.6	33.3	98.4	66.7	93.8
	SRF (OCT only) SRF (OCT + FA + ICGA)	N/A	77.0	21.2	95.1	58.3	78.9
		N/A	80.7	24.2	99.0	88.9	80.2
	PED (OCT only) PED (OCT + FA + ICGA)	N/A	77.0	70.4	94.6	97.2	54.7
		N/A	80.7	74.5	97.3	98.7	59.0
	SHRM (OCT only) SHRM (OCT + FA + ICGA)	N/A	91.9	76.5	94.1	65.0	96.5
		N/A	96.3	88.2	97.5	83.3	98.3
Zhao et al. [21]—efficacy of anti-VEGF injection on post treatment BCVA at approx. 28 days by classyifing responders and non-responders	Responders/non- responders	0.83	84.6	0.692	1	N/A	N/A

**Table 16 diagnostics-13-02464-t016:** Machine learning and deep learning models for predicting conversion to nAMD and the datasets used for prediction.

Authors	Date	Model	Dataset
**Machine Learning**			
Schmidt-Erfurth et al. [13]	July 2018	Cox proportional hazard	495 iAMD fellow eyes of 495 nAMD patients undergoing anti-VEGF treatment within the HARBOR clinical trial
**Deep Learning**			
Russakoff et al. [22]	February 2019	AMDnet and VGG16	71 early or iAMD fellow eyes of 71 nAMD patients undergoing anti-VEGF treatment at Moorfields Eye Hospital
Banerjee et al. [24]	September 2020	Deep Sequence LSTM	671 early or iAMD fellow eyes from 671 nAMD patients undergoing anti-VEGF treatment with the HARBOR clinical trial
		Deep Sequence LSTM	719 early or iAMD eyes from 507 patients visiting Bascom Palmer Eye Institute (MIAMI) between 2004 to 2015
Yim et al. [23]	May 2020	Ensemble DL model	2261 non-nAMD fellow eyes from 2795 nAMD patients visiting seven sites of Moorfields Eye Hospital between 2012 and 2017

**Table 17 diagnostics-13-02464-t017:** Machine learning and deep learning models for predicting conversion to nAMD with the prediction tasks, scenarios, numbers of visits, prediction time frame and results (BL = baseline; C = conservative operating point; CPH = Cox proportional hazard; L = liberal operating point; m = months; Se. = sensitivity; Sp. = specificity; TA = total available; V = visits).

Author and Prediction Task	Model and Scenario	Number of Visits	Prediction Time Frame	AUC	Se.	Sp.
Schmidt-Erfurth et al. [13]—risk for conversion from iAMD to nAMD	CPH	BL + 4 V	24 m	0.68	0.80	0.46
Russakoff et al. [22]—risk for conversion from early or iAMD to nAMD	AMDNet—OCT B-scan	BL	24 m	0.89	N/A	N/A
	AMDnet—OCT volume	BL	24 m	0.91	N/A	N/A
	VGG16—OCT B-scan	BL	24 m	0.82	N/A	N/A
	VGG16—OCT volume	BL	24 m	0.87	N/A	N/A
Banerjee et al. [24]—risk for conversion from early or iAMD to nAMD	Deep Sequence LSTM (Harbor)	TA	3 m	0.96	N/A	N/A
		TA	6 m	0.83	N/A	N/A
		TA	9 m	0.78	N/A	N/A
		TA	12 m	0.77	N/A	N/A
		TA	15 m	0.84	N/A	N/A
		TA	18 m	0.90	N/A	N/A
		TA	21 m	0.97	N/A	N/A
		BL + 4 V	18 m	N/A	0.88	0.87
	Deep Sequence LSTM (Miami)	TA	3 m	0.82	N/A	N/A
		TA	6 m	0.77	N/A	N/A
		TA	9 m	0.71	N/A	N/A
		TA	12 m	0.69	N/A	N/A
		TA	15 m	0.68	N/A	N/A
		TA	18 m	0.65	N/A	N/A
		TA	21 m	0.68	N/A	N/A
		BL + 4 V	18 m	N/A	0.68	0.71
Yim et al. [23]—risk for conversion from non-nAMD to nAMD	Ensemble DL model—OCT volume (conversion scan)	One	6 m	0.745	0.80 (L); 0.34 (C)	0.55 (L); 0.90 (C)
	Ensemble DL model—OCT volume (injection scan)	One	6 m	0.884	0.80 (L); 0.34 (C)	0.55 (L); 0.90 (C)

**Table 18 diagnostics-13-02464-t018:** Potential biases for studies predicting the conversion to nAMD (green = lower risk for bias, red = higher risk for bias; a sample under 100 patients was considered small; RWD = real-world data; RCT = randomized clinical trial; RCLP = real-life clinical practice; FA = fluorescein angiography).

Authors	Small Sample Size	Data Source	External Validation	Automatically Extracted Features	Quality of Conversion Interpretation
Schmidt-Erfurth et al. [13]	No	RCT	No	Yes	RCT OCT monitoring
Russakoff et al. [22]	Yes	RWD	No	N/A	RLCP OCT and FA monitoring
Banerjee et al. [24]	No	RCT	Yes	Yes	RCT OCT monitoring
Yim et al. [23]	No	RWD	No	Yes	RLCP OCT monitoring and injection start

**Table 19 diagnostics-13-02464-t019:** Machine learning and deep learning models for predicting conversion to GA with their datasets.

Authors	Date	Dataset	Model
**Machine Learning**			
Schmidt-Erfurth et al. [13]	July 2018	495 iAMD fellow eyes of 495 nAMD patients undergoing anti-VEGF treatment within the HARBOR clinical trial	Cox proportional hazard
**Deep Learning**			
Rivail et al. [25]	October 2019	420 iAMD eyes from 221 patients	Deep Siamese network

**Table 20 diagnostics-13-02464-t020:** Machine learning and deep learning models for predicting conversion to GA (AUC = area under the curve; BL = baseline; CPH = Cox proportional hazard; m = months; N/A = not available; P = precision; Se = sensitivity; Sp = specificity; V = visits.

Author and Prediction	Model	No. of Visits	Time Frame	AUC	Se	Sp	P
**Machine Learning**							
Schmidt-Erfurth et al. [13]—risk for conversion from iAMD to GA	CPH	BL + 4 V	24 m	0.80	0.80	0.69	N/A
**Deep Learning**							
Rivail et al. [25]—risk for conversion from iAMD to GA	Deep Siamese network	One	6 m	0.753	N/A	N/A	0.367
		One	12 m	0.784	N/A	N/A	0.394
		One	18 m	0.773	N/A	N/A	0.463

**Table 21 diagnostics-13-02464-t021:** Potential biases for studies predicting the conversion to GA (green = lower risk for bias, red = higher risk for bias, yellow = unknown/unclear risk for bias; a sample under 100 patients was considered small; RCT = randomized clinical trial; N/A = not applicable).

Authors	Small Sample Size	Data Source	External Validation	Automatically Extracted Features	Quality of Conversion Interpretation
Schmidt-Erfurth et al. [13]	No	RCT	No	Yes	RCT OCT monitoring
Rivail et al. [25]	No	Unknown	No	N/A	Unknown

**Table 22 diagnostics-13-02464-t022:** Machine learning and deep learning models and the datasets used for predicting GA growth.

Authors	Date	Model	Dataset
**Machine Learning**			
Niu et al. [26]	June 2016	Random forest classifier	38 GA eyes from 29 patients visiting Byers Eye Institute of Stanford University
**Deep Learning**			
Zhang et al. [27]	February 2021	BiLSTM + 3D-UNet	22 GA eyes from 22 patients visiting Byers Eye Institute of Stanford University and 3 GA eyes from 3 patients visiting Jiangsu Provincial People’s Hospital in China
Gigon et al. [28]	November 2021	Encoder–decoder CNN + time-based Taylor series	129 GA eyes from 119 patients who visited the Jules Gonin Eye Hospital in Lausanne, Switzerland

**Table 23 diagnostics-13-02464-t023:** Machine learning and deep learning prediction scenarios for GA growth. The DI was calculated for overall future GA lesions (* mean DI for overall GA lesions) (BiLSTM = bi-directional long short-term memory; DI = Dice index; GT = ground truth).

Scenario	Predicted GA Growth	Training Data	DI
**Niu et al. [26]—Random Forest Classifier**			
1	First follow-up visit for each patient	BL + first follow-up from all other patients	0.81 *
2	All consecutive follow-up visits for each patient	BL + first follow-up from all other patients	0.84 *
3	Second and subsequent follow-up visits for each patient	BL + first follow-up scan from the same patient	0.87 *
**Zhang et al. [27]—BiLSTM + 3D-UNet**			
1	All consecutive follow-up visits for each patient	BL + first two follow-up visits from all other patients	0.86 *
2	All consecutive follow-up visits for each patient	All consecutive follow-up visits from all other patients	0.89 *
3	Third and subsequent follow-up visits for each patient	BL + first two follow-up visits from the same patient	0.89 *
4	Last follow-up visit for each patient	All prior consecutive follow-up visits from the same patient	0.92 *
5	Third and subsequent follow-up visits for all patients	BL + first two follow-up visits from all patients	0.88 *
6	Last follow-up visit for all patients	All prior consecutive follow-up visits from all patients	0.90 *
**Gigon et al. [28]—Encoder–Decoder CNN + Time-based Taylor Series**			
A	BL and subsequent follow-up visits	BL visit—manually segmented GT	0.73 to 0.80 *
A	BL and subsequent follow-up visits	BL visit—automatically segmented GT	0.74 to 0.81 *
B	Next follow-up visit	Preceding visit—manually segmented GT	0.83 to 0.88 *
B	Next follow-up visit	Preceding visit—automatically segmented GT	0.84 to 0.89 *

**Table 24 diagnostics-13-02464-t024:** Potential biases for studies predicting GA growth (green = lower risk for bias, red = higher risk for bias; a sample under 100 patients was considered low; RWD = real-world Data; GT = ground truth; N/A = not applicable).

Authors	Small Sample Size	Data Source	External Validation	Automatically Extracted Features	Quality of GA Growth GT
Niu et al. [26]	Yes	RWD	No	Yes	Automatic segmentation with visual review
Zhang et al. [27]	Yes	RWD	No	N/A	Automatic segmentation with visual review
Gigon et al. [28]	No	RWD	No	Yes	Automatic and manual segmentation

## Data Availability

Not applicable.

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
