# Peer review of "The Predictive Capabilities of Artificial Intelligence-Based OCT Analysis for Age-Related Macular Degeneration Progression—A Systematic Review"

_diagnostics, 2023, doi:10.3390/diagnostics13142464_

Round 1
Reviewer 1 Report
Thank you for conducting this comprehensive systematic review on the predictive capabilities of Artificial Intelligence-based OCT analysis for Age-related Macular Degeneration (AMD) progression. Your paper provides valuable insights into the potential of AI in improving the accuracy and efficiency of diagnosis and monitoring of AMD. (1) However, it would be beneficial if you could address the limitations associated with the reviewed studies. Identifying and discussing any potential biases, such as small sample sizes or variations in study design, would strengthen the overall conclusions of your review. (2) Highlighting the specific challenges faced by AI-based OCT analysis in real-world clinical settings, such as variations in imaging devices or data quality, could further enhance the practical implications of your findings.(3) "PRISMA" should be introduced to the reader so that they know what it is and what it entails. (4) Sections 2.1, 2.4.1. need improvement. (5) You need to indicate whether Figure 3 is scaled and should be added as a caption. (6) The contents of Table 1 need to be centered so that it can be easily read. (7) There are some excessive spaces within the article, adjust it. (8) Some abbreviations are not defined; double-check the point.
Minor editing of English language required
Reviewer 2 Report
- The article demonstrates a effort on the part of the authors to carry out the systematic review. However, the article needs to be greatly improved in order to be accepted.
- The article suggests (although it does not explicitly specify) the date range considered, from 2012 to 2022. The number of articles retrieved in that date range was approximately 1400. I am not sure if the search was really exhaustive, or just few articles have been published in this area. For the first case (incomplete search) it is recommended to review the queries. For the second case, it is suggested to wait for a better moment to carry out a systematic review.
- One of the main problems in the design of this systematic review is that the authors did not start with the research questions that would guide the process.
- What were the keywords for the search?
Other minor observations are the following:
- Please, use institutional emails instead of generic ones (gmail, yahoo, etc.)
- The use of punctuation marks is suggested, in order to express ideas in shorter sentences.
- Therefore, the article cannot be accepted for publication.
- The use of punctuation marks is suggested, in order to express ideas in shorter sentences.
Round 2
Reviewer 1 Report
Thanks to authors for carefully addressing my comments. The revised version is much improved and looks in a better to position to be accepted.
Therefore, my recommendation is to accept it.
Minor editing of English language required